# Gaussian Process Bandits for Top-k Recommendations

**Mohit Yadav**
University of Massachusetts Amherst
`ymohit@cs.umass.edu`

**Daniel Sheldon**
University of Massachusetts Amherst
`sheldon@cs.umass.edu`

**Cameron Musco**
University of Massachusetts Amherst
`cmusco@cs.umass.edu`

## Abstract

Algorithms that utilize bandit feedback to optimize top-k recommendations are vital for online marketplaces, search engines, and content platforms. However, the combinatorial nature of this problem poses a significant challenge, as the possible number of ordered top-k recommendations from $n$ items grows exponentially with $k$. As a result, previous work often relies on restrictive assumptions about the reward or bandit feedback models, such as assuming that the feedback discloses rewards for each recommended item rather than a single scalar feedback for the entire set of top-k recommendations. We introduce a novel contextual bandit algorithm for top-k recommendations, leveraging a Gaussian process with a Kendall kernel to model the reward function. Our algorithm requires only scalar feedback from the top-k recommendations and does not impose restrictive assumptions on the reward structure. Theoretical analysis confirms that the proposed algorithm achieves sub-linear regret in relation to the number of rounds and arms. Additionally, empirical results using a bandit simulator demonstrate that the proposed algorithm outperforms other baselines across various scenarios.

## 1   Introduction

The top-$k$ recommendation problem involves providing a ranked list of k items, such as news articles or products, from a pool of $n$ items [34, 13]. Online algorithms must adapt to dynamic user preferences, making bandit algorithms suitable due to their use of limited feedback [1]. Developing bandit algorithms is challenging due to limited feedback and the need for computational efficiency in real-time recommendation environments. Recent research on user interfaces for recommendations highlights that the overall layout of the recommendation page is crucial for user appeal, as modern UI designs have evolved from simple dropdown lists to complex, visually engaging layouts [17, 13, 18]. Consequently, bandit algorithms must jointly select and display all top-$k$ items, rather than simply choosing the most relevant $k$ items and ordering them by decreasing user relevance [31].

The joint consideration of top-$k$ items makes the number of arms (possible actions for the bandit algorithm) combinatorially large, i.e., $\Theta(n^k)$. Previous research on bandit algorithms often imposes strict assumptions on feedback models [30, 21]. For instance, *semi-bandit* feedback provides a scalar reward for each of the top $k$ items, thus decomposing the combinatorial feedback into item-level feedback. However, this type of feedback is frequently unavailable [32]. Another common feedback model is *cascade* browsing [16], which assumes that users examine items in a predetermined order and cease browsing once a desirable item is found, offering item-specific scalar feedback but failing to capture potential non-linear interactions among items [26]. Figure 1 illustrates the limitations of the cascade model in capturing user interactions within modern top-$k$ recommendation interfaces.

38th Conference on Neural Information Processing Systems (NeurIPS 2024).

These limitations motivate us to adopt a more general *full-bandit* feedback setting, where only a single scalar value is provided for the entire top-$k$ set of recommendations [24].

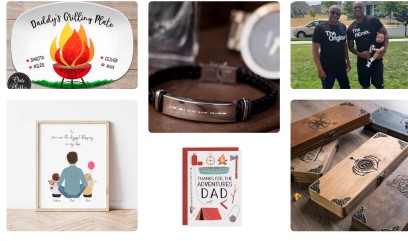

**Figure 1:** A snapshot from Etsy showcases Father's Day shopping recommendations. The lack of an obvious linear search order challenges the assumptions of the cascade model. Additionally, the proximity and arrangement of items are likely to influence clicks, indicating complex interaction patterns and supporting the need for full-bandit feedback without assumptions about user interactions with recommended items.

**Table 1:** Compute and memory analysis for the proposed GP-TopK bandit algorithm. Rows represent different costs: total *compute* and *memory* of the GP-TopK algorithm for $\mathbf{T}$ rounds, time for matrix-vector multiplication (**mvm**) with the kernel matrix $K_{X_t}$ for $t^{\text{th}}$ round, and time to update $K_{X_t}$. Columns represent different approaches: the *kernel approach*, which uses full kernel matrices, and our novel *feature approach*, which performs the same operations through feature expansions and scales more efficiently with respect to $\mathbf{T}$. The symbols $c$, $k$, and $\mathbf{T}$ denote the embedding size for contexts, the number of items, and the number of rounds, respectively.

| Tasks | kernel approach | feature approach |
|---|---|---|
| *compute* | $\mathcal{O}(\mathbf{T^3})$ | $\mathcal{O}(c \cdot k^2 \cdot \mathbf{T^2})$ |
| *memory* | $\mathcal{O}(\mathbf{T^2})$ | $\mathcal{O}(c \cdot k^2 \cdot \mathbf{T})$ |
| **mvm**$(K_{X_t})$ | $\mathcal{O}(t^2)$ | $\mathcal{O}(c \cdot k^2 \cdot t)$ |
| *compute* $K_{X_t}$ | $\mathcal{O}((c + k^2) \cdot t)$ | $\mathcal{O}(c \cdot k^2)$ |

Beyond feedback assumptions, the reward structure in bandit algorithms must be decomposable into scalar values for individual items to prevent a combinatorial explosion of arms—something that is not always feasible. For example, modern e-commerce platforms value and track metrics such as diversity and fairness [1], which cannot be captured by focusing solely on individual items [15]. This necessitates algorithms for full-bandit feedback settings that operate without specific assumptions about the objectives or reward structures [24].

This work introduces a bandit algorithm that uses Gaussian processes (GPs) to model rewards under full-bandit feedback (i.e., a single scalar value). GPs are selected for their flexibility in accommodating feedback across discrete, continuous, and mixed domains, such as continuous contexts and discrete rankings [33]. Additionally, unlike parametric models that require optimization incorporating accumulated feedback from previous rounds, GP updates are computationally efficient, involving only data updates [24]. Although GP inference may face computational limits, we will develop efficient inference methods tailored to our proposed algorithm. A further challenge in designing GP-based bandit algorithms for top-$k$ recommendations is constructing expressive positive-definite kernels that capture similarities between top-$k$ recommendations [9]. This work mitigates these computational and modeling challenges, as illustrated in the following sections.

Broadly speaking, GPs have been previously explored for bandit algorithms [27, 19]. Krause et al. [14] employed GPs for contextual bandits in continuous domains; we focus on the discrete domain of top-$k$ recommendations. Vanchinathan et al. [28] used GPs with a position-based feedback model, and Wang et al. [31] used GPs with semi-bandit feedback for recommending top-$k$ items. In contrast, our work does not rely on a specific reward model or feedback assumption, and develops an an efficient GP-based bandit algorithm for top-$k$ recommendations.

## 1.1 Contributions

Our primary contribution is the GP-TopK algorithm, a contextual bandit algorithm for recommending top-$k$ items. This algorithm operates in a full-bandit feedback setting without relying on assumptions on reward, making it broadly applicable compared to prior works. We leverage GPs with variants of the *Kendall* kernel [12] to model the reward function and optimize the upper confidence bound (UCB) [27] acquisition function for selecting the next arm. Additionally, we introduce a novel weighted convolutional Kendall kernel for top-$k$ recommendations that address pathologies in existing variants of the Kendall kernel when applied to top-$k$ recommendations.

Our second key contribution is enhancing the scalability of the GP-TopK algorithm for longer time horizons. Initially, the computational cost for top-$k$ recommendations using the GP-TopK algorithm is $\mathcal{O}(T^4)$ for $T$ rounds. We first reduce this to $\mathcal{O}(T^3)$ by leveraging iterative algorithms from numerical linear algebra [25]. Next, we derive sparse feature representations for the novel weighted

convolutional Kendall kernel, further reducing the compute requirements from $\mathcal{O}(T^4)$ to $\mathcal{O}(T^2)$ and memory requirements from $\mathcal{O}(T^2)$ to $\mathcal{O}(T)$. Table 1 summarizes these improvements in time and memory requirement, including their dependence on other parameters.

We provide a theoretical analysis showing that GP-TopK's regret is sub-linear in $T$, benefiting from the feature representations of the Kendall kernels introduced in this work. Specifically, we establish an upper bound on regret that is nearly quadratic in $n$, significantly improving over the naive $\Theta(n^k)$ bound for top-$k$ recommendations without using feature representations [27]. Finally, we empirically validate GP-TopK's regret on real-world datasets, demonstrating improvement over baseline methods.

## 1.2 Organization

The remainder of this paper is organized as follows: Section 2 introduces Kendall kernels for full and top-$k$ rankings, including the novel weighted convolutional Kendall kernel. Section 3 presents faster matrix-vector multiplication algorithms for Kendall kernels, enhancing the efficiency of the proposed bandit algorithm, which is further detailed along with the regret analysis in Section 4. Finally, Sections 5 and 6 present empirical results and concluding discussion, respectively.

## 2 Kendall Kernels for Full and Top-k Rankings

This section introduces Kendall kernels and their extensions for top-$k$ recommendations, forming the foundation of our approach. We first establish key notations and them present Sections 2.1 and 2.2, which introduce Kendall kernels for full rankings and top-$k$ rankings, respectively.

**Notations:** Let $[n] = 1, 2, \ldots, n$, with $\pi$ representing a top-$k$ ranking—an ordered tuple of $k$ distinct elements from $[n]$. For a full ranking ($k = n$), we use $\sigma$ and denote the set of all possible top-$k$ rankings by $\Pi^k$, with cardinality $|\Pi^k| = \Theta(n^k)$. To capture ranking positions, the vector $\mathbf{p}^\sigma \in \mathbb{R}^n$ corresponds to a full ranking $\sigma$ with entry $\mathbf{p}_i^\sigma$ gives the rank of item $i$. For top-k rankings, $\mathbf{p}^\pi \in \mathbb{R}^n$ is similarly constructed by arbitrarily assigning distinct ranks to items not in the top $k$. For relative ranks, indicator functions $\mathfrak{p}_{i<j}^\sigma$ and $\mathfrak{p}_{i>j}^\sigma$ denote whether item $i$ is ranked before or after item $j$, respectively in $\sigma$. Also, $\mathfrak{p}_{i<j}^\pi$ and $\mathfrak{p}_{i>j}^\pi$ are similar indicator functions defined for top-k rankings.

### 2.1 Kendall Kernels for Full Rankings

Jiao et al. [9] showed that the Kendall tau rank correlation coefficient [12] is a positive definite (p.d.) kernel for full rankings, which we refer to as the standard Kendall (SK) kernel. The weighted Kendall (WK) kernel generalizes the SK kernel by differentially weighting item pairs [10]. Specifically, the SK and WK kernels for full rankings $\sigma_1, \sigma_2$ are defined as:

$$k^{sk}(\sigma_1, \sigma_2) := \frac{1}{\binom{n}{2}} \sum_{i<j} \eta_{i,j}(\sigma_1, \sigma_2) \tag{1}$$

$$k^{wk}(\sigma_1, \sigma_2) := \frac{1}{\binom{n}{2}} \sum_{i<j} w((\mathbf{p}_i^{\sigma_1}, \mathbf{p}_j^{\sigma_1}), (\mathbf{p}_i^{\sigma_2}, \mathbf{p}_j^{\sigma_2})) \cdot \eta_{i,j}(\sigma_1, \sigma_2), \tag{2}$$

where $\eta_{i,j}$ is 1 if the pair $(i, j)$ is *concordant* (ordered the same in both rankings) and $-1$ otherwise; concretely, $\eta_{i,j}(\sigma_1, \sigma_2) := \mathfrak{p}_{i<j}^{\sigma_1} \cdot \mathfrak{p}_{i<j}^{\sigma_2} + \mathfrak{p}_{i>j}^{\sigma_1} \cdot \mathfrak{p}_{i>j}^{\sigma_2} - \mathfrak{p}_{i<j}^{\sigma_1} \cdot \mathfrak{p}_{i>j}^{\sigma_2} - \mathfrak{p}_{i>j}^{\sigma_1} \cdot \mathfrak{p}_{i<j}^{\sigma_2}$; and $w((\mathbf{p}_i^{\sigma_1}, \mathbf{p}_j^{\sigma_1}), (\mathbf{p}_i^{\sigma_2}, \mathbf{p}_j^{\sigma_2}))$ is the value of a positive definite weighting kernel $w(\cdot, \cdot) : [n]^2 \times [n]^2 \mapsto \mathbb{R}$ that operates on pairs of ranks. The $w_{i,j}$ adds flexibility and can assign varying importance to ranks, similar to the discounted cumulative gain (DCG) metric [7]. Note that both SK and WK kernels are p.d. and right-invariant with respect to $\Pi^n$ [10]. In other words, they compute similarity based only on the relative ranks of pairs, not on the labels of items, as clearly evident from Equations 1 and 2.

### 2.2 Kendall Kernels for Top-k Rankings

**Weighted Kendall and Convolutional Kendall (CK) kernels.** To adapt the WK kernel from full rankings to top-$k$ rankings, Jiao et al. [10] set the weighting function $w(i, j, \sigma_1, \sigma_2)$ to zero if either item is not in the top-k of either ranking. While this approach yields a p.d. kernel, it disregards

items outside the intersection of top-$k$ rankings. In contrast, the convolutional operation provides an alternative for adapting the standard Kendall kernel to top-$k$ rankings.

Let $B_\pi$ denote the set of full rankings consistent with the top-k ranking $\pi$ (i.e., for every item $i$ in $\pi$, $\forall \sigma \in B_\pi, \mathbf{p}_i^\pi = \mathbf{p}_i^\sigma$). The convolutional Kendall kernel can be defined as follows:

$$k^{ck}(\pi_1, \pi_2) = \frac{1}{|B_{\pi_1}| \cdot |B_{\pi_2}|} \sum_{\sigma_1 \in B_{\pi_1}, \ \sigma_2 \in B_{\pi_2}} k^{sk}(\sigma_1, \sigma_2), \tag{3}$$

where $k^{sk}$ is the standard Kendall kernel. Since the CK kernel is a convolution of another p.d. kernel, it is also a p.d. kernel [5]. Unlike the WK kernel for top-k rankings, the CK kernel accounts for items not in both top-k rankings. However, computing the CK kernel is expensive, requiring exponentially many evaluations of the kernel $k^{sk}$ in the double summation. Therefore, Jiao et al. [9] developed an efficient algorithm to bypass this double summation, reducing compute to $\mathcal{O}(k \log k)$ time.

**Proposed Weighted Convolutional Kendall (WCK) Kernel.** To combine the strengths of the WK and CK kernels for top-k rankings, we propose the weighted convolutional Kendall kernel for top-k rankings $\pi_1$ and $\pi_2 \in \Pi^k$:

$$k^{wck}(\pi_1, \pi_2) := \frac{1}{|B_{\pi_1}| \cdot |B_{\pi_2}|} \sum_{\sigma_1 \in B_{\pi_1}, \sigma_2 \in B_{\pi_2}} k^{wk}(\sigma_1, \sigma_2), \tag{4}$$

where $k^{wk}$ represents the weighted Kendall kernel for full rankings $\sigma_1, \sigma_2 \in \Pi^n$.

The proposed WCK kernel combines the flexibility of differentially weighting ranks among the top-$k$ items (as in the WK kernel) with the ability to account for items outside the intersection of both top-$k$ rankings (as in the CK kernel). Additionally, as a convolution of a p.d. kernel, it is also a p.d. kernel. However, computing the WCK kernel remains challenging, as it requires exponentially many evaluation of the $k^{wk}$ kernel, as given in the RHS of Equation 4. To address this, we focus on a specific form of rank weights of the $k^{wk}$ kernel, called as *product-symmetric* rank weights:

$$w_{ps}((i_1, j_1), (i_2, j_2)) := w_s(i_1, j_1) \cdot w_s(i_2, j_2), \tag{5}$$

where, $w_s(i, j) : [n] \times [n] \mapsto \mathbb{R}$ is a symmetric function, i.e., $w_s(i, j) = w_s(j, i)$. Notably, the WCK kernel can be computed efficiently for the case of these weights (see Claim 1 below).

The WCK kernel, even with the relatively simple $w_{ps}$ weights, exhibits notable properties, as shown in Table 2. In this table, we use $w_s(i, j) = \frac{1}{\log(i+1)} \cdot \frac{1}{\log(j+1)}$, inspired by the DCG metric commonly applied in recommendation systems [7]. Notably, the WK kernel ranks two rankings with no overlap ($\pi_0$ and $\pi_1$) as more similar than two rankings with the same items in reversed order ($\pi_0$ and $\pi_2$), indicating a clear pathology. Further, the CK kernel fails to distinguish between reversed pairs at different ranks ($k^{ck}(\pi_0, \pi_3) = k^{ck}(\pi_0, \pi_4)$), presenting another limitation if known variants of Kendall kernels for top-$k$ rankings. By using product-symmetric ranking weights, the WCK kernel addresses these shortcomings, providing a more nuanced similarity comparison for top-$k$ rankings.

**Table 2:** Comparison of Kendall kernel similarities for top-k rankings. The table shows kernel values $k(\pi_0, \cdot)$ for the top-k ranking $\pi_0 = [1, 2, 3]$ with other rankings ($\pi_1$, $\pi_2$, $\pi_3$, $\pi_4$) for $n = 7$ and $k = 3$. Rankings are arranged left to right by increasing similarity to $\pi_0$. The similarity values provided by the proposed kernel increase from left to right as expected, demonstrating the desirable behavior of the WCK kernel with DCG rank weights, unlike other variants. All kernels are unit-normalized. See text for further details.

| Top-k Kernels | $\pi_1$ [4, 5, 6] | $\pi_2$ [3, 2, 1] | $\pi_3$ [2, 1, 3] | $\pi_4$ [1, 3, 2] |
|---|---|---|---|---|
| WK | 0.00 | −1.00 | 0.33 | 0.33 |
| CK | −0.60 | 0.60 | 0.87 | 0.87 |
| WCK | −0.38 | 0.09 | 0.46 | 0.87 |

> **Claim 1.** *The weighted convolutional Kendall kernel (Equation 4) with product-symmetric rank weights (Equation 5) can be computed in $\mathcal{O}(k^2)$ time.*

Appendix A provides the proof that leverages the structure of product-symmetric rank weights $w_{ps}$ to establish the existence of a feature representation for the WCK kernel, as formally stated below in

Claim 3 below. We then demonstrate that the inner product of these features, and hence the WCK kernel, can be computed in $\mathcal{O}(k^2)$ time (Algorithm 2 in the appendix). Similar to the result of Jiao et al. [9] for the CK kernel, this approach avoid exponentially many evaluations of $k^{wk}$ on the RHS of Equation (4) by enabling a direct computation of the WCK kernel.

## 3  Fast Matrix-Vector Multiplication with Kendall Kernel Matrices

In Gaussian processes, inference can be accelerated by using iterative algorithms that take advantage of fast matrix-vector-multiplications (MVMs) with the kernel matrix [3]. This section introduces fast algorithms for kernel MVMs by exploiting the implicit structure of Kendall kernel matrices.

Let $\mathbf{mvm}(K_{X_t})$ denote the runtime required to multiply the $t \times t$ kernel matrix $K_{X_t} = (k(x_i, x_j))_{x_i, x_j \in X_t}$ by any admissible vector. In the naive approach, this runtime is $\mathbf{mvm}(K\tilde{X}_t) = \mathcal{O}(t^2)$. However, if $k(x_i, x_j) = \phi^a(x_i)^T \phi^b(x_j)$ for any arbitrary $x_i$ and $x_j$, where the vectors $\phi^a(x_i)$ and $\phi^b(x_j)$ are sparse and contain only $z$ non-zero entries, then $\mathbf{mvm}(K_{X_t})$ reduces to $\mathcal{O}(z \cdot t)$, which is a significant improvement over $\mathcal{O}(t^2)$ when $z \ll t$. When $\phi^a = \phi^b$, we refer to $\phi^a$ as the *linear feature vector* for the kernel $k$. Before focusing on top-k ranking kernels, we provide a linear feature vector for the WK kernel on full rankings (given earlier in Equation 2).

> **Claim 2.** *Let $\phi^{wk}(\sigma) : \Pi^n \mapsto \mathbb{R}^{\binom{n}{2}}$ be a vector indexed by unique item pairs $(i, j)$, defined as:*
>
> $$\phi^{wk}_{i,j}(\sigma) := \frac{1}{\sqrt{\binom{n}{2}}} \cdot w_s(\mathbf{p}^\sigma_i, \mathbf{p}^\sigma_j) \cdot \left( \mathfrak{p}^\sigma_{i<j} - \mathfrak{p}^\sigma_{i>j} \right),$$
>
> *where $w_s$ is the symmetric weighting function in product-symmetric weights. Then, $\phi^{wk}$ is a linear feature vector for the weighted Kendall kernel with product-symmetric weights $w_{ps}$.*

Using Claim 2, the linear feature vector for the WK kernel can be extended to the WK top-$k$ ranking kernel by utilizing the structure of product-symmetric weights, which allows weights to be set to zero for items outside of the top-$k$ rankings, as described in Section 2.2. Precisely, such a feature vector for the top-$k$ ranking kernel is sparse; specifically, the feature vector $\phi^{wk}(\pi)$ contains only $\mathcal{O}(k^2)$ non-zero entries due to the WK kernel's focus on item pairs within the top-$k$. Consequently, the runtime for $\mathbf{mvm}(K_{X_t})$ in the WK kernel matrix is reduced to $\mathcal{O}(k^2 \cdot t)$.

Moving forward, we focus on deriving a sparse feature vector for the WCK kernel, enabling fast MVMs with the WCK kernel, which includes the CK kernel as a special case. Notably, any convolutional kernel inherits linear features from its constituent kernel. Specifically, $\sum_{\sigma \in B_\pi} \phi^{wk}_{i,j}(\sigma)$ forms a feature vector for the WCK kernel, which follows from Equation 4 and However, computing this feature vector explicitly is computationally challenging, as it requires summing over all $\sigma \in B_\pi$, which includes an exponential number of terms, i.e., $\Theta(n^k)$.

In response to this challenge, Claim 3 shows that the summation can be computed analytically and provides explicit linear feature vectors for the WCK and CK kernels. It also shows that $\phi^{wck}$ has only $\mathcal{O}(k^2 + 2nk)$ non-zero entries among its $\mathcal{O}(n^2)$ total entries. Consequently, $\mathbf{mvm}(K_{X_t})$ for

> **Claim 3.** *Let $\phi^{wck}(\pi) : \Pi^k \mapsto \mathbb{R}^{\binom{n}{2}}$ be a vector indexed by unique item pairs $(i, j)$ given as: $\phi^{wck}_{i,j}(\pi) := \frac{1}{\sqrt{\binom{n}{2}}} \cdot \mathbf{w}^{wck}_{i,j}(\pi) \cdot \left( \mathfrak{p}^\pi_{i<j} - \mathfrak{p}^\pi_{i>j} \right)$, where $\mathbf{w}^{wck}_{i,j}(\pi)$ is determined as follows:*
>
> $$\mathbf{w}^{wck}_{i,j}(\pi) = \begin{cases} w_s(\mathbf{p}^\pi_i, \mathbf{p}^\pi_j) & \text{if } \mathbf{p}^\pi_i \in [k] \ \& \ \mathbf{p}^\pi_j \in [k] \\ w_s(\mathbf{p}^\pi_i, \cdot) & \text{else if } \mathbf{p}^\pi_i \in [k] \ \& \ \mathbf{p}^\pi_j \notin [k], \\ w_s(\mathbf{p}^\pi_j, \cdot) & \text{else if } \mathbf{p}^\pi_i \notin [k] \ \& \ \mathbf{p}^\pi_j \in [k], \\ 0 & \text{otherwise}, \end{cases}$$
>
> *where $w_s$ denotes symmetric weights and $w_s(\ell, \cdot) = \frac{1}{n-k} \sum_{j=k+1}^n w_s(\ell, j)$. Then, the vector $\phi^{wck}$ is a linear feature vector for the WCK kernel $k^{wck}$. By uniformly setting $w_s(\cdot, \cdot) \equiv 1$ in the definitions above, $\phi^{wck}_{i,j}(\pi)$ specializes to a linear feature vector for the CK kernel.*

the WCK kernel requires $\mathcal{O}((k^2 + 2nk) \cdot t)$ operations, which improves from $\mathcal{O}(t^2)$ to linear in $t$. However, this introduces a dependence on $n$, the number of items, which poses a serious limitation and is beneficial only when $n \leq t$. In the following theorem, we leverage redundancy in $\phi^{wck}$ to eliminate this dependence on $n$, leading to the following main theorem about the $\mathbf{mvm}(K_{X_t})$.

**Theorem 1.** *For the WCK kernel with product-symmetric weights $w_{ps}$, the computational complexity of multiplying the kernel matrix $K_{X_t}$ with any admissible vector is $\mathcal{O}(k^2 t)$, i.e., $\mathbf{mvm}(K_{X_t}) = \mathcal{O}(k^2 t)$, where $X_t$ is any arbitrary set of $t$ top-k rankings.*

Appendix A provides the proof in two steps. First, we utilize the values of $\phi^{wck}$ from Claim 3 and categorize $\phi^{wck}(\pi_1)^T \phi^{wck}(\pi_2)$ based on item pairs, as summarized in Table 4. Next, we show that only five combinations yield non-zero values, i.e., $\phi^{wck}(\pi_1)^T \phi^{wck}(\pi_2) = \sum_{i=1}^{5} s_i(\pi_1, \pi_2)$. Each term $s_i(\pi_1, \pi_2)$ is a dot product of vectors $\phi^{a_i}(\pi_1)^T \phi^{b_i}(\pi_2)$, which contains at most $\mathcal{O}(k^2)$ non-zero entries. Thus, for the WCK and CK kernels, $\mathbf{mvm}(K_{X_t}) = \mathcal{O}(k^2 t)$, since these vectors across all five terms include only $\mathcal{O}(k^2)$ non-zero entries. Consequently, Theorem 1 demonstrates that employing these vector representations for top-k rankings leads to faster MVMs, i.e., $\mathbf{mvm}(K_{X_t}) = \mathcal{O}(k^2 t) \ll \mathcal{O}(t^2)$.

## 4  Proposed GP-TopK Bandit Algorithm

In this section, we begin by formally defining the top-$k$ recommendation problem within a bandit framework and introduce a generic contextual bandit algorithm, detailed in Algorithm 1. We then explain how the components of the algorithm are instantiated using the proposed GP approach, followed by an analysis of its computational complexity and cumulative regret.

Let $T$ denote the number of rounds. Contexts $\mathcal{C}$ are represented in a finite $c$-dimensional space, i.e., $\mathcal{C} \subseteq \mathbb{R}^c$. In the $t^{th}$ round, we receive a context $\mathbf{c}_t \in \mathcal{C}$ and select a top-$k$ ranking $\pi_t \in \Pi^k$. Subsequently, a noisy reward $y_t = \hat{f}(\mathbf{c}_t, \pi_t) + \epsilon_t$ is observed, where $\hat{f}$ is the true reward function and $\epsilon_t$ is round-independent noise. The regret is defined as $r_t := \max \pi^{'} \in \Pi^k \hat{f}(\mathbf{c}_t, \pi^{'}) - \hat{f}(\mathbf{c}_t, \pi_t)$, with cumulative regret $R_T := \sum_{t=1}^{T} r_t$. The accumulated data at the $t^{th}$ round is $\mathcal{D}_t = (\mathbf{c}_i, \pi_i, y_i)_{i=1}^{t}$. Below, the Algorithm 1 provides provides a generic schematic of the bandit algorithm.

---

**Algorithm 1** Contextual Bandit Algorithm for Top-k Recommendations

---
**Input:** Total rounds $T$, initial reward model $\mathcal{M}_0$, and acquisition function $\mathcal{AF}$.
 1: **for** $t = 1, \cdots, T$ **do**
 2:    Observe a context $\mathbf{c}_t$ from the context space $\mathcal{C}$.
 3:    Select a top-$k$ ranking $\pi_t$ that maximizes $\mathcal{AF}(\mathcal{M}_{t-1}(\mathbf{c}_t, \pi))$ for the context $\mathbf{c}_t$.
 4:    Obtain the scalar reward $y_t$.
 5:    Update the reward model $\mathcal{M}_t$ using the accumulated feedback $\mathcal{D}_t$.
 6: **end for**

---

We aim to design the components of above Algorithm 1 with the objectives of minimizing cumulative regret and ensuring computational efficiency. It requires two key components: (a) a reward model $\mathcal{M}_t$ that estimates the reward for any context and top-$k$ ranking utilizing the accumulated data $\mathcal{D}_t$ and (b) an acquisition function $\mathcal{AF}$ for selecting $\pi_t$ given the reward model $\mathcal{M}_t$ and observed context $\mathbf{c}_t$.

**Reward model $\mathcal{M}$ and acquisition function $\mathcal{AF}$.**  The proposed GP-TopK bandit algorithm leverages GP regression to model the reward function over the domain of contexts and top-k rankings. Section B.1 briefs GP regression for the completeness. Essentially, the reward model $\mathcal{M}$ maintains a distribution over functions $f$, i.e., $f \sim \mathcal{N}(0, k(\cdot, \cdot))$, where $k$ is a product kernel function over both contexts and top-k rankings ($\mathcal{C} \bigotimes \Pi^k$). Specifically, the kernel function $k$ is defined as follows:

$$k((\mathbf{c}_1, \pi_1), (\mathbf{c}_2, \pi_2)) := k^c(\mathbf{c}_1, \mathbf{c}_2) \cdot k^r(\pi_1, \pi_2), \tag{6}$$

where $k^c(\mathbf{c}_1, \mathbf{c}_2) = \mathbf{c}_1^T \mathbf{c}_2$ is the dot-product kernel and $k^r$ is a kernel for top-k rankings. We use variants of the Kendall kernel for $k^r$ from Section 2. Updating the reward model $\mathcal{M}_t$ at the $t^{\text{th}}$

round involves adding new data points to our GP regression, which is computationally inexpensive compared to the fine-tuning steps required by parametric models to incorporate the latest feedback.

We use the UCB function as the acquisition function, balancing exploration and exploitation by selecting actions that maximize the upper confidence bound of the estimated reward [27]. The UCB acquisition function is $\mathcal{AF}(\mathcal{M}_t(\mathbf{c}_t, \pi)) := \mu_{f|\mathcal{D}}((\mathbf{c}_t, \pi)) + \beta^{\frac{1}{2}} \cdot \sigma_{f|\mathcal{D}}((\mathbf{c}_t, \pi))$, where $\sigma_{f|\mathcal{D}}((\mathbf{c}_t, \pi)) = \sqrt{k_{f|\mathcal{D}}((\mathbf{c}_t, \pi), (\mathbf{c}_t, \pi))}$ and $\beta$ controls the trade-off between exploration and exploitation. Here, $\mu_{f|\mathcal{D}}$ and $k_{f|\mathcal{D}}$ are the GP posterior mean and covariance functions, as detailed in Section B.1. At the $t^{\text{th}}$ round, the algorithm selects the top-k ranking $\pi \in \Pi^k$ that maximizes $\mathcal{AF}(\mathcal{M}_t(\mathbf{c}_t, \pi))$, which is performed using local search [19], as detailed further in Appendix B.

**Computational complexity.** The GP-TopK bandit algorithm does not require compute for model updates. In other words, updating $\mathcal{M}_t$, i.e., in the Line 5 of the Algorithm 1 requires only updating the list of accumulated feedback data $\mathcal{D}_t$. The GP-TopK relies on local search to optimize $\mathcal{AF}$, so the computational demands stem solely from $\mathcal{AF}$ evaluations within the local search. As shown in Section B.1, computing the GP variance term for evaluating $\mathcal{AF}$, i.e, $\sigma_{f|\mathcal{D}}((\mathbf{c}_t, \pi))$ involves solving $\left[K_{X_t} + \sigma^2 I\right]^{-1} \mathbf{v}$ for a vector $\mathbf{v}$, where $X_t = [(\mathbf{c}_1, \pi_1), \cdots, (\mathbf{c}_t, \pi_t)]$. Naively, this operation requires $\mathcal{O}(t^3)$ time per round, amounting to total $\mathcal{O}(T^4)$ over $T$ rounds. Iterative algorithms, however, can expedite the process by leveraging fast MVMs with kernel matrices, as discussed in Section 3. Below, Theorem 2 formalizes the computational demands of the GP-TopK algorithm.

> **Theorem 2.** *Assuming a fixed number of iterations required by the iterative algorithms, the total computational time for running the GP-TopK bandit algorithm for $T$ rounds of top-k recommendations, using the contextual product kernel (Equation 6), is $\mathcal{O}(k^2 c \ell T^2)$. This applies to WK, CK, and WCK top-k ranking kernels, where $\ell$ is the number of local search evaluations.*

The proof of Theorem 2, provided in Appendix B, demonstrates efficiency gains from combining feature representations with iterative algorithms, reducing computational time from $\mathcal{O}(T^4)$ to $\mathcal{O}(T^2)$. This is a substantial improvement, as even a single MVM with the matrix $K_{X_t}$ using the full kernel matrix at each round would require $\mathcal{O}(T^3)$ compute time. Additionally, the theorem shows that the running time of the GP-TopK algorithm does not explicitly depend on the number of items $n$.

**Regret analysis.** The cumulative regret is $R_T = \sum_{t=1}^{T} \max_{\pi' \in \Pi^k} \hat{f}(\mathbf{c}_t, \pi') - \hat{f}(\mathbf{c}_t, \pi_t)$, where $\pi_t$ is the ranking chosen at round $t$. Optimizing cumulative regret for top-$k$ recommendations is challenging, as it requires learning the context-arm relationship and matching the best possible mapping. To bound cumulative regret, regularity assumptions are essential, as noted in prior works [27, 14]. *We consider the following two assumptions, either of which suffices.* Also, $\mathcal{X} := \mathcal{C} \bigotimes \Pi^k$ for below assumptions.

**Assumption 1.** $\mathcal{X}$ *is finite, meaning that only finite contexts are considered ($|\mathcal{C}| < \infty$), and the reward function $\hat{f}$ is sampled from the GP prior with a noise variance of $\xi^2$.*

**Assumption 2.** $\mathcal{X}$ *is arbitrary and the reward function $\hat{f}$ has a bounded RKHS norm for the kernel $k$, i.e., $\|f\|_k \leq B$. The reward noises $\epsilon_t$ form an arbitrary martingale difference sequence (i.e., reward noise does not systematically depend on its past values) and are uniformly bounded by $\xi$.*

The following theorem proves the regret bound for the GP-TopK algorithm under Assumption 1 or 2.

> **Theorem 3.** *If either Assumptions 1 or 2 hold, setting $\beta_t$ as $2\log\left(\frac{|\mathcal{C}| \cdot |\Pi^k| \cdot t^2 \cdot \pi^2}{6\delta}\right)$ and $300\gamma_t \ln^3\left(\frac{t}{\delta}\right)$ respectively, the cumulative regret $\mathcal{R}_T$ of the GP-TopK bandit algorithm for top-k recommendations can, with at least $1 - \delta$ probability, be bounded by $\tilde{\mathcal{O}}(n\sqrt{C_1 T c (\log|\mathcal{C}| + k + \log(T^2 \pi^2/6\delta))})$ under Assumption 1, and $\tilde{\mathcal{O}}(n\sqrt{C_1(2B^2c + 300n^2c^2\ln^3(T/\delta))T})$ under Assumption 2. Here, $C_1 = \frac{8}{\log(1+\xi^{-2})}$, and $\tilde{\mathcal{O}}$ excludes logarithmic factors related to $n$, $k$, and $T$.*

Appendix B.4 provides the proof, leveraging the insight that $\log \det |I + \xi^{-2} \cdot K_{X_T}|$ for any set $X_T$ can be effectively bounded using the finite-dimensional feature vectors introduced in this work.

Specifically, Proposition 2 utilizes the feature vectors from Section 2. Building on Proposition 2, Theorem 3 establishes that the cumulative regret of the GP-TopK bandit algorithm grows sublinearly in $T$ with high probability for both assumptions. Furthermore, this result also underscore the importance of using top-k ranking kernels, which improve the asymptotic order in terms of $n$ by factors of $n^{k/2-1}$ and $n^{k-1}$ under Assumptions 1 and 2, respectively, compared to Srinivas et al. [27]. This improvement is substantial even for small values of $k$, such as $k = 6$, as shown in Table 3.

**Table 3:** Comparison with Srinivas et al. (2010) on regret bounds for the bandit algorithm under both assumptions. Definitions of notations are provided in the main text.

| Assumption 1 | |
|:---:|:---:|
| **Srinivas et al. (2010)** | **Proposed GP-TopK Algorithm** |
| $\tilde{O}\left(n^{\frac{k}{2}}\sqrt{C_1 T c \left(\log\|\mathcal{C}\| + k + \log\left(\frac{T^2\pi^2}{6\delta}\right)\right)}\right)$ | $\tilde{O}\left(n\sqrt{C_1 T c \left(\log\|\mathcal{C}\| + k + \log\left(\frac{T^2\pi^2}{6\delta}\right)\right)}\right)$ |
| **Assumption 2** | |
| **Srinivas et al. (2010)** | **This work** |
| $\tilde{O}\left(n^{\frac{k}{2}}\sqrt{C_1 T c \left(2B^2 + 300n^k c \ln^3\left(\frac{T}{\delta}\right)\right)}\right)$ | $\tilde{O}\left(n\sqrt{C_1 T c \left(2B^2 + 300n^2 c \ln^3\left(\frac{T}{\delta}\right)\right)}\right)$ |

## 5 Experiments

This section empirically evaluates the proposed GP-TopK bandit algorithms for the top-k recommendations using a simulation based on the MovieLens dataset [4]. The reliance on simulation for evaluating bandit algorithms is prevalent in the literature. It stems from the difficulty of conducting online evaluations in real-world bandit scenarios, mainly when there are combinatorial arms [28]. Next, we provide details of the simulation setup and considered reward settings. Following that, we present results for the empirical regret for small and large numbers of arms below, respectively.

**Simulation setup and reward settings.** The bandit simulation setup follows the framework outlined by Jeunen et al. [8], utilizing real-world datasets on user-item interactions. Specifically, we train user and item embeddings using a collaborative filtering approach [6]. The user embeddings are accessed by the bandit algorithms as context embeddings, while the item embeddings remain hidden. In the non-contextual setup, the first user from the dataset is chosen as a fixed context throughout the bandit algorithm run, allowing us to use the same reward functions as the contextual bandit algorithm.

For setting up the reward functions, we utilize a similarity function $s(\mathbf{c}, \theta) \coloneqq \varsigma(a \cdot (\mathbf{c}^T\theta) - b)$ to measure similarity between any user and item embeddings, where $a$ and $b$ are similarity score and shift scalars, respectively. The sigmoid function $\varsigma$ maps similarity scores to a range between 0 and 1, enhancing the interpretability of the reward signal [31]. We set $a$ and $b$ to 6 and 0.3, respectively, to fully utilize the range of the similarity function, as assessed by evaluating its value for many arms.

We set up two preliminary reward functions based on the similarity function $s$. The first is the DCG metric, $\hat{f}_{\text{dcg}}(\mathbf{c}, \pi) = \sum_{i=1}^{k} \frac{1}{\log_2(i+1)} s(\mathbf{c}, \theta_{\pi_i})$, where $\mathbf{c}$ and $\theta_{\pi_i}$ represent the context and item embeddings, respectively. The second is the diversity measure, $\hat{f}_{\text{div}}(\pi) = \frac{1}{k^2} \sum_{i=1}^{k} \sum_{j=1}^{k} \theta_{\pi_j}^T \theta_{\pi_i}$. These metrics quantify the relevance and diversity of top-k recommendations, respectively.

We use these functions in two contextual reward settings. The first setting focuses on normalized-DCG (n-DCG), $\hat{f}_{\text{ndcg}}(\mathbf{c}, \pi) = \frac{\hat{f}_{\text{dcg}}(\mathbf{c}, \pi)}{\max_{\pi'} \hat{f}_{\text{dcg}}(\mathbf{c}, \pi')}$ [7]. The second setting combines $\hat{f}_{\text{ndcg}}$ and $\hat{f}_{\text{div}}$ as $\hat{f}_{\text{ndcgdiv}}(\mathbf{c}, \pi) = \lambda \cdot \hat{f}_{\text{ndcg}}(\mathbf{c}, \pi) + (1 - \lambda) \cdot \hat{f}_{\text{div}}(\pi)$, evaluating the aggregate effect of relevance and diversity. We set $\lambda = 0.75$ to emphasize relevance over diversity.

**Evaluation for small arm space.** This section presents empirical results for the cumulative regret of bandit algorithms with a limited number of arms. Specifically, with $n = 20$ and $k = 3$, there are $6,840$ top-k rankings, allowing for an exhaustive search to optimize the acquisition function. All bandit algorithms run in batch mode, updating every five rounds. We consider both reward settings for contextual and non-contextual scenarios, using a subset of five users for the contextual setting.

Several baselines are set to assess the benefits of ranking (Kendall) kernels. Section C details the remaining hyper-parameter configurations and details of other baseline bandit algorithms.

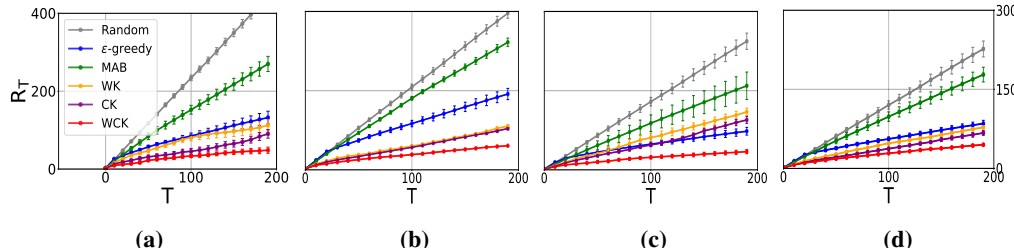

|     (a)     |     (b)     |     (c)     |     (d)     |

**Figure 2:** Comparative evaluation of bandit algorithms: The cumulative regret $R_T$ over $T$ rounds is shown. Lower values indicate better performance. Plots (a) and (b) represent non-contextual settings for nDCG ($\hat{f}_{\text{ndcg}}$) and nDCG + diversity ($\hat{f}_{\text{ndcgdiv}}$) rewards, respectively. Plots (c) and (d) show results for contextual settings for five users using the same rewards. The y-axis for (a) and (b) is on the left, and for (c) and (d) on the right. The GP-TopK algorithm with Kendall kernels, especially the weighted convolutional Kendall (WCK) kernel, outperforms others. Details on other algorithms are in the text. Results are averaged over six trials.

The *Random* algorithm randomly recommends any k items. The $\epsilon$-*greedy* algorithm alternates between recommending a random top-k ranking with a probability of $\epsilon$ and choosing the top-k ranking with the highest observed mean reward. In contextual settings, $\epsilon$-*greedy* differentiates arms for each unique context. Similarly, *MAB-UCB* conceptualizes each ranking as an independent arm, an equivalent of using a direct delta kernel approach for GPs along with UCB $\mathcal{AF}$. In contextual scenarios, *MAB-UCB* also treats arms distinctly per context. Each variant of the top-k ranking kernel yields one variation of the proposed GP-TopK algorithm, namely, WK, CK, and WCK. Figure 2 presents empirical values of the cumulative regrets for the above baseline and the proposed GP-TopK algorithms. In all cases, across both reward settings and in both contextual and non-contextual setups, the variants of the proposed GP-TopK algorithm outperform baselines that do not use Kendall kernels, highlighting the significance of top-k ranking kernels for full bandit feedback. Specifically, the CK and WCK kernels significantly outperform the WK kernel regarding the converged values of the regret, with the WCK kernel further improving on the CK kernel variant.

**Evaluation for large arm space.**
We evaluate bandit algorithms in a large arm space scenario with $n = 50$ and $k = 3$ and $k = 5$, resulting in $1.1 \times 10^5$ and $1.1 \times 10^{10}$ possible top-k rankings, respectively. Using local search, we focus on the nDCG reward. The remaining configuration is consistent with the small arm space setup. We use 10 restarts and 5 steps in each search direction for the local search, starting with 1000 initial candidates.

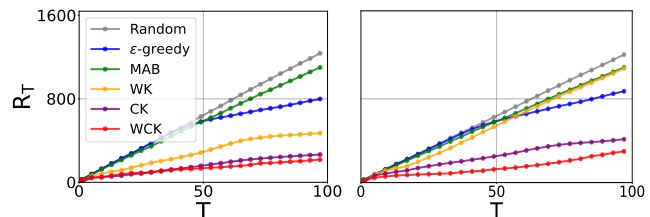

**Figure 3:** Comparative evaluation of bandit algorithms for large arm spaces, with $> 1.1 \times 10^5$ arms for the left plot and $> 1.1 \times 10^{10}$ arms for the right plot. Cumulative regret with respect to the rounds of the bandit algorithm is depicted. Results are averaged over six trials. In both settings, the WCK approach outperforms other baselines. For more details, see the textual description.

Figure 3 shows that the regret for the GP-TopK variants remains consistently lower even with a large arm space, despite the use of local search. The WCK approach significantly outperforms the CK, especially for $k = 5$, as illustrated in the right plot of Figure 3. Additional empirical results on the effectiveness of local search in a large arm space and other rewards are given in Appendix C.

## 6   Discussion

This work develops a contextual bandit algorithm for top-$k$ recommendations using Gaussian processes with Kendall kernels in a full-bandit feedback setting, without restrictive assumptions about feedback or reward models. Gaussian processes provide computationally efficient model updates

for accumulated feedback data, although inference can be challenging. We address this by deriving features for Kendall kernels tailored to top-$k$ rankings, resulting in a faster inference algorithm that reduces complexity from $\mathcal{O}(T^4)$ to $\mathcal{O}(T^2)$. While demonstrated here for the product kernel between contexts and top-$k$ rankings, these computational improvements extend naturally to other kernel types, such as additive kernels. Additionally, we address limitations of known variants and propose a more expressive Kendall kernel for top-$k$ recommendations. Finally, we provide both theoretical and empirical results demonstrating the improved performance of the proposed GP-TopK algorithm.

**Future Directions and Limitations.**    This work opens several research avenues. Efficient matrix-vector multiplication with Kendall kernel matrices can enable faster bandit algorithms with various acquisition functions, such as Thompson sampling and expected improvement. Exploring other kernels, like Mallow kernels, for top-k rankings and developing efficient algorithms for them is an intriguing direction, especially since the effectiveness of our algorithm depends on the function space induced by the RKHS of the underlying kernel. Assessing how well these kernels approximate various reward functions for top-k recommendations would provide valuable insights.

Exploring other bandit problem settings, such as stochastic item availability or delayed feedback, would enhance the applicability of this work to more complex scenarios. Extending the finite-dimensional GP framework to other acquisition functions using local search is another promising direction. One limitation of our regret analysis is that it does not account for approximations in the arm selection step due to local search [20]. This limitation is common in continuous domains, where optimizing acquisition functions often involves non-convex optimization [27].

**Impact.**    This research advances bandit algorithms for top-k item recommendations. By improving recommendation efficiency and accuracy, our algorithms can enhance user experiences across platforms, promoting content relevancy and engagement. However, they may reinforce implicit biases in training data, limiting content diversity and entrenching prejudices. Therefore, monitoring over time is essential when deploying these algorithms in real-world environments.

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

## A  Kendall Kernels for Full and Top-k Rankings – Omitted Details

This section includes the proofs that were omitted from Section 2, presented in the following order:

- In Section A.1, we present proofs for Claims 2 and 3, which concern the feature representations of Kendall kernels.

- In Section A.2, we provide Algorithms 2 and a proof of its correctness for computing the WCK kernel in $\mathcal{O}(k^2)$ time, thereby proving Claim 1. Additionally, we extend this proof to cover the proof of correctness for Algorithm 3, which can compute the CK kernel in $\mathcal{O}(k \log k)$, initially introduced by Jiao et al. [9]. The original paper presented the algorithm without a formal proof of correctness, a gap we address and fill in this section.

- Section A.3 details the proof for Theorem 1, discussing the matrix-vector multiplications with the Kendall kernel matrix for top-k rankings. This proof builds on the Algorithm 2 given for computing the WCK kernel for top-k rankings.

### A.1  Feature Representation for Kendall Kernels for Top-k Rankings

This section revisits the claims regarding the feature representations of the weighted Kendall kernel and the weighted convolutional Kendall kernel, subsequently providing the proofs for these claims mentioned earlier.

---

**Claim 2.** *Let $\phi^{wk}(\sigma) : \Pi^n \mapsto \mathbb{R}^{\binom{n}{2}}$ be a vector indexed by unique item pairs $(i, j)$, defined as:*

$$\phi_{i,j}^{wk}(\sigma) := \frac{1}{\sqrt{\binom{n}{2}}} \cdot w_s(\mathbf{p}_i^\sigma, \mathbf{p}_j^\sigma) \cdot \left(\mathfrak{p}_{i<j}^\sigma - \mathfrak{p}_{i>j}^\sigma\right),$$

*where $w_s$ is the symmetric weighting function in product-symmetric weights. Then, $\phi^{wk}$ is a linear feature vector for the weighted Kendall kernel with product-symmetric weights $w_{ps}$.*

---

*Proof.* Following the definition of linear feature representation, we need to prove that $k^{wk}(\sigma_1, \sigma_2) = \phi(\sigma_1)^T \phi(\sigma_2)$ for the product-symmetric weight kernel as given in Equation 5. Recalling from Equation 2, we have $k^{wk}(\sigma_1, \sigma_2)$ as follows:

$$k^{wk}(\sigma_1, \sigma_2) = \frac{1}{\binom{n}{2}} \cdot \sum_{i<j} w((\mathbf{p}_i^{\sigma_1}, \mathbf{p}_j^{\sigma_1}), (\mathbf{p}_i^{\sigma_2}, \mathbf{p}_j^{\sigma_2})) \cdot \eta_{i,j}(\sigma_1, \sigma_2),$$

$$= \frac{1}{\binom{n}{2}} \cdot \sum_{i<j} w_s(\mathbf{p}_i^{\sigma_1}, \mathbf{p}_j^{\sigma_1}) \cdot w_s(\mathbf{p}_i^{\sigma_2}, \mathbf{p}_j^{\sigma_2}) \cdot \eta_{i,j}(\sigma_1, \sigma_2), \qquad (7)$$

where the second line incorporates the use of the product-symmetric weight kernel. Next, our focus shifts to the simplification of $\eta_{i,j}(\sigma_1, \sigma_2)$, which is elaborated as follows:

$$\eta_{i,j}(\sigma_1, \sigma_2) = \mathfrak{p}_{i<j}^{\sigma_1} \cdot \mathfrak{p}_{i<j}^{\sigma_2} + \mathfrak{p}_{i>j}^{\sigma_1} \cdot \mathfrak{p}_{i>j}^{\sigma_2} - \mathfrak{p}_{i<j}^{\sigma_1} \cdot \mathfrak{p}_{i>j}^{\sigma_2} - \mathfrak{p}_{i>j}^{\sigma_1} \cdot \mathfrak{p}_{i<j}^{\sigma_2},$$

$$= \mathfrak{p}_{i<j}^{\sigma_1} \cdot (\mathfrak{p}_{i<j}^{\sigma_2} - \mathfrak{p}_{i>j}^{\sigma_2}) + \mathfrak{p}_{i>j}^{\sigma_1} \cdot (\mathfrak{p}_{i>j}^{\sigma_2} - \mathfrak{p}_{i<j}^{\sigma_2}),$$

$$= (\mathfrak{p}_{i<j}^{\sigma_1} - \mathfrak{p}_{i>j}^{\sigma_1}) \cdot (\mathfrak{p}_{i<j}^{\sigma_2} - \mathfrak{p}_{i>j}^{\sigma_2}).$$

Combining the above factorization of $\eta_{i,j}$ with Equation 7, we get:

$$k^{wk}(\sigma_1, \sigma_2) = \frac{1}{\binom{n}{2}} \cdot \sum_{i<j} w_s(\mathbf{p}_i^{\sigma_1}, \mathbf{p}_j^{\sigma_1}) \cdot w_s(\mathbf{p}_i^{\sigma_2}, \mathbf{p}_j^{\sigma_2}) \cdot (\mathfrak{p}_{i<j}^{\sigma_1} - \mathfrak{p}_{i>j}^{\sigma_1}) \cdot (\mathfrak{p}_{i<j}^{\sigma_2} - \mathfrak{p}_{i>j}^{\sigma_2})$$

$$= \frac{1}{\binom{n}{2}} \cdot \sum_{i<j} \phi_{i,j}^{wk}(\sigma_1) \cdot \phi_{i,j}^{wk}(\sigma_2)$$

$$= \phi(\sigma_1)^T \phi(\sigma_2).$$

$\square$

---

**Claim 3.** *Let $\phi^{wck}(\pi) : \Pi^k \mapsto \mathbb{R}^{\binom{n}{2}}$ be a vector indexed by unique item pairs $(i,j)$ given as: $\phi_{i,j}^{wck}(\pi) := \frac{1}{\sqrt{\binom{n}{2}}} \cdot \mathbf{w}_{i,j}^{wck}(\pi) \cdot \left(\mathfrak{p}_{i<j}^\pi - \mathfrak{p}_{i>j}^\pi\right)$, where $\mathbf{w}_{i,j}^{wck}(\pi)$ is determined as follows:*

$$\mathbf{w}_{i,j}^{wck}(\pi) = \begin{cases} w_s(\mathbf{p}_i^\pi, \mathbf{p}_j^\pi) & \text{if } \mathbf{p}_i^\pi \in [k] \ \& \ \mathbf{p}_j^\pi \in [k] \\ w_s(\mathbf{p}_i^\pi, \cdot) & \text{else if } \mathbf{p}_i^\pi \in [k] \ \& \ \mathbf{p}_j^\pi \notin [k], \\ w_s(\mathbf{p}_j^\pi, \cdot) & \text{else if } \mathbf{p}_i^\pi \notin [k] \ \& \ \mathbf{p}_j^\pi \in [k], \\ 0 & \text{otherwise,} \end{cases}$$

*where $w_s$ denotes symmetric weights and $w_s(\ell, \cdot) = \frac{1}{n-k} \sum_{j=k+1}^n w_s(\ell, j)$. Then, the vector $\phi^{wck}$ is a linear feature vector for the WCK kernel $k^{wck}$. By uniformly setting $w_s(\cdot, \cdot) \equiv 1$ in the definitions above, $\phi_{i,j}^{wck}(\pi)$ specializes to a linear feature vector for the CK kernel.*

---

*Proof.* The main idea revolves around leveraging the feature representation of the Weighted Kendall kernel for a full ranking and the linearity of the convolution operation. It is already established that $k^{wk}(\sigma_1, \sigma_2) = \phi^{wk}(\sigma_1)^T \phi^{wk}(\sigma_2)$, as demonstrated in Claim 2. Recall that the WCK kernel requires a double summation over pairs of rankings from $B_{\pi_1}$ and $B_{\pi_2}$, which represent the sets of full rankings consistent with their respective top-k rankings, as described in Equation 4. We simplify the WCK kernel as follows:

$$k^{wck}(\pi_1, \pi_2) = \frac{1}{|B_{\pi_1}|} \cdot \frac{1}{|B_{\pi_2}|} \cdot \sum_{\sigma_1 \in B_{\pi_1}} \sum_{\sigma_2 \in B_{\pi_2}} \phi^{wk}(\sigma_1)^T \phi^{wk}(\sigma_2)$$

$$= \left( \frac{1}{|B_{\pi_1}|} \cdot \sum_{\sigma_1 \in B_{\pi_1}} \phi^{wk}(\sigma_1)^T \right) \cdot \underbrace{\left( \frac{1}{|B_{\pi_2}|} \cdot \sum_{\sigma_2 \in B_{\pi_2}} \phi^{wk}(\sigma_2) \right)}_{:=\phi^{wck}(\pi_2)}$$

$$= \phi^{wck}(\pi_1)^T \phi^{wck}(\pi_2).$$

The simplification above reveals that the feature representation, $\phi^{wck}$, for the WCK kernel, is a $\binom{n}{2}$ dimensional vector and can be indexed by unique pairs of items $(i,j)$, much like the $\phi^{wk}$. However, the double summation is over an exponentially large number of pairs of rankings. Moving forward, we shift our focus to the individual entries of this representation involving this summation, elucidating the analytical values within the summation by exploring four unique cases, each dependent on whether these specific items fall within the top-k rankings.

In Case 1, we examine the scenario where items $i$ and $j$ are within the top-k ranking $\pi$. Here, the focus is on the feature representation of the pair, specifically when both elements are ranked among the top-k positions.

Case 1: $\mathbf{p}_i^\pi \in [k]$ and $\mathbf{p}_j^\pi \in [k]$.

$$\phi_{i,j}^{wck}(\pi) = \frac{1}{|B_\pi|} \cdot \sum_{\sigma \in B_\pi} \frac{1}{\sqrt{\binom{n}{2}}} \cdot w_s(\mathbf{p}_i^\sigma, \mathbf{p}_j^\sigma) \cdot \left(\mathfrak{p}_{i<j}^\sigma - \mathfrak{p}_{i>j}^\sigma\right)$$

$$= \frac{1}{|B_\pi|} \cdot \frac{1}{\sqrt{\binom{n}{2}}} \cdot w_s(\mathbf{p}_i^\pi, \mathbf{p}_j^\pi) \cdot \left( \sum_{\sigma \in B_\pi} \mathfrak{p}_{i<j}^\sigma - \sum_{\sigma \in B_\pi} \mathfrak{p}_{i>j}^\sigma \right)$$

$$= \frac{1}{|B_\pi|} \cdot \frac{1}{\sqrt{\binom{n}{2}}} \cdot w_s(\mathbf{p}_i^\pi, \mathbf{p}_j^\pi) \cdot \left( |B_\pi| \cdot \mathfrak{p}_{i<j}^\pi - |B_\pi| \cdot \mathfrak{p}_{i>j}^\pi \right)$$

$$= \frac{1}{\sqrt{\binom{n}{2}}} \cdot w_s(\mathbf{p}_i^\pi, \mathbf{p}_j^\pi) \cdot \left( \mathfrak{p}_{i<j}^\pi - \mathfrak{p}_{i>j}^\pi \right). \tag{8}$$

The simplification in lines 3rd and 4th follows from the fact that any full ranking $\sigma \in B_\pi$, consistent with the top-k ranking $\pi$, the relative ranks and weights of items $i$ and $j$ remains unchanged, given $\mathbf{p}_i^\pi \in [k]$ and $\mathbf{p}_j^\pi \in [k]$. Concretely, this implies $\mathfrak{p}_{i<j}^\sigma = \mathfrak{p}_{i<j}^\pi$ for all $\sigma \in B_\pi$ and similar with the other term.

In Case 2, we analyze when item $i$ is in the top-k ranking while item $j$ is not.

Case 2: $\mathbf{p}_i^\pi \in [k]$ and $\mathbf{p}_j^\pi \notin [k]$.

$$\phi_{i,j}^{wck}(\pi) = \frac{1}{|B_\pi|} \cdot \sum_{\sigma \in B_\pi} \frac{1}{\sqrt{\binom{n}{2}}} \cdot w_s(\mathbf{p}_i^\sigma, \mathbf{p}_j^\sigma) \cdot \left( \mathfrak{p}_{i<j}^\sigma - \mathfrak{p}_{i>j}^\sigma \right)$$

$$= \frac{1}{|B_\pi|} \cdot \frac{1}{\sqrt{\binom{n}{2}}} \cdot \sum_{\sigma \in B_\pi} w_s(\mathbf{p}_i^\sigma, \mathbf{p}_j^\sigma) \cdot (1 - 0) \ \ (\text{since } \mathbf{p}_i^\pi \in [k] \text{ and } \mathbf{p}_j^\pi \notin [k])$$

$$= \frac{1}{|B_\pi|} \cdot \frac{1}{\sqrt{\binom{n}{2}}} \cdot \sum_{\sigma \in B_\pi} w_s(\mathbf{p}_i^\pi, \mathbf{p}_j^\sigma).$$

Next, every possible consistent ranking is considered jointly while fixating on a specific rank outside top-k elements, leading to $(n - k - 1)!$ different rankings. Given that $|B_\pi| = (n - k)!$, we can refine the above expression as follows:

$$\phi_{i,j}^{wck}(\pi) = \frac{1}{|B_\pi|} \cdot \frac{1}{\sqrt{\binom{n}{2}}} \cdot \sum_{l=k+1}^{n} w_s(\mathbf{p}_i^\pi, l) \cdot (n - k - 1)!$$

$$= \frac{(n - k - 1)!}{|B_\pi|} \cdot \frac{1}{\sqrt{\binom{n}{2}}} \cdot \sum_{l=k+1}^{n} w_s(\mathbf{p}_i^\pi, l)$$

$$= \frac{1}{\sqrt{\binom{n}{2}}} \cdot \frac{1}{n - k} \cdot \sum_{l=k+1}^{n} w_s(\mathbf{p}_i^\pi, l)$$

$$= \frac{1}{\sqrt{\binom{n}{2}}} \cdot w_s(\mathbf{p}_i^\pi, \cdot). \tag{9}$$

In Case 3, we analyze when item $i$ is not in the top-k ranking while item $j$ is.

Case 3: $\mathbf{p}_i^\pi \notin [k]$ and $\mathbf{p}_j^\pi \in [k]$. Similar to case 2, the simplification follows analogously, with the only change being $\mathbf{1}_{\mathbf{p}_{i<j}^\sigma} - \mathbf{1}_{\mathbf{p}_{i>j}^\sigma} = -1$ instead of 1. Thus, by symmetry between $i$ and $j$, we have the following:

$$\phi_{i,j}^{wck}(\pi) = \frac{-1}{\sqrt{\binom{n}{2}}} \cdot w_s(\cdot, \mathbf{p}_j^\pi) = \frac{-1}{\sqrt{\binom{n}{2}}} \cdot w_s(\mathbf{p}_j^\pi, \cdot) \qquad (\text{using symmetry of } w_s). \tag{10}$$

Lastly, in Case 4, we analyze when items $i$ and $j$ are not in the top-k ranking.

Case 4: $\mathbf{p}_i^\sigma \notin [k]$ and $\mathbf{p}_j^\sigma \notin [k]$.

$$\phi_{i,j}^{wck}(\pi) = \frac{1}{|B_\pi|} \cdot \sum_{\sigma \in B_\pi} \phi_{i,j}^{wk}(\sigma)$$

$$= \frac{1}{|B_\pi|} \cdot \frac{1}{\sqrt{\binom{n}{2}}} \cdot \sum_{\sigma \in B_\pi} w_s(\mathbf{p}_i^\sigma, \mathbf{p}_j^\sigma) \cdot \left( \mathfrak{p}_{i<j}^\sigma - \mathfrak{p}_{i>j}^\sigma \right)$$

$$= 0 \quad \text{(by symmetry).} \tag{11}$$

The result of zero arises from symmetry. Since $\mathbf{p}_i^\sigma$ and $\mathbf{p}_j^\sigma$ are not in the top-$k$ ranking, they are treated symmetrically in the summation overall rankings in $B_\pi$. For any ranking $\sigma$, suppose $\mathbf{p}_i^\sigma = l$ and $\mathbf{p}_j^\sigma = m$, there exists a corresponding ranking $\sigma'$ such that only the items $i$ and $j$ are swapped. Therefore, jointly, these two rankings yield $w_s(l, m)$ and $-w_s(l, m)$. Since $w_s$ is symmetric, the overall contribution from each pair of such rankings is zero. Hence, the entire summation nets to zero.

Thus, with the explanation provided for each case and combining results from Equations 8, 9, 10 and 11, it's trivial to validate the Claim 3, i.e., $\phi_{i,j}^{wck}(\pi) = \frac{1}{\sqrt{\binom{n}{2}}} \cdot \mathbf{w}_{i,j}^{wck}(\pi) \cdot \left( \mathfrak{p}_{i<j}^\pi - \mathfrak{p}_{i>j}^\pi \right)$ for all unique pair of items. From Case 4, we have $\mathcal{O}((n-k)^2)$ entries leaving at max only $\mathcal{O}(k^2 + 2nk)$ non-zero entries. $\qquad \square$

## A.2 Algorithms for Computing Kendall Kernels for top-k Rankings

In this section, we provide and delve into the proofs of Algorithms 2 and 3 for the weighted convolutional Kendall kernel and the convolutional Kendall kernel, as previously discussed in Section 2. Section A.2.1 for valid both the correctness and computational complexity of Algorithm 2 as given earlier in Claim 1. Following this, Section A.2.2 revisits Algorithm 3, initially introduced by Jiao et al. [10]. The original publication presented the algorithm without formal proof of its correctness, which we rectify and offer in Section A.2.2.

### A.2.1 Efficiently Computing the Weighted Convolutional Kendall Kernel

This section provides a proof to Claim 1 to establish the efficiency and accuracy of Algorithm 2 in computing the weighted convolutional Kendall kernel, as specified in Equation 4, with a focus on its computational complexity.

> **Claim 1.** *The weighted convolutional Kendall kernel (Equation 4) with product-symmetric rank weights (Equation 5) can be computed in $\mathcal{O}(k^2)$ time.*

*Proof.* The claim is proven through Algorithm 2, where we establish its correctness and demonstrate its computation requirement is $O(k^2)$. The essence of our proof centers on analyzing the feature representation of the WCK kernel, $\phi^{wck}$, as outlined in Claim 3. The feature vectors of $\phi^{wck}$ reside in a $\binom{n}{2}$ dimensional space, indexed by pairs of items. Our approach is to demonstrate that Algorithm 2 accurately computes the right-hand side (RHS) of the equation $k^{wck}(\pi_1, \pi_2) = \phi^{wck}(\pi_1)^T \phi^{wck}(\pi_2)$. This involves a summation over item pairs, expressed as $k^{wck}(\pi_1, \pi_2) = \sum_{l<m} \phi_{l,m}^{wck}(\pi_1)^T \phi_{l,m}^{wck}(\pi_2)$.

Our proof analyzes various scenarios: cases where pairs of items, namely $l$ and $m$, fall within the top-k, scenarios with one item within the top-k and the other outside, and situations where neither item is within the top-k. Each of these cases contributes distinctively to the computation of the overall kernel, resulting in different terms in the algorithmic computation. This is encapsulated in Algorithm 2, where $k^{wck}(\pi_1, \pi_2) = \sum_{i=1}^{5} s_i(\pi_1, \pi_2)$, and each $s_i$ corresponds to the terms given earlier in Algorithm 2 from Section 2.

Before proceeding with the cases of this summation as given in Table 4, we recall the notations utilized by Algorithm 3 in Definition 1. Also, remember that we will be proving for product-symmetric weights as given in Equation 5, where, $w_s : [n] \times [n] \mapsto \mathbb{R}^n$ and its one-dimensional marginals are $w_s(\ell, \cdot) = \frac{1}{n-k} \sum_{j=k+1}^{n} w_s(\ell, j)$ Table 4 shows how these cases are organized and relate to different $s_i$ terms required for computing the WCK kernel. The key strategy involves breaking down the kernel's computation into cases based on the positioning of item pairs within the top-k rankings. In case 1, we consider all the scenarios when both indices are within the set of items in both top-k rankings, i.e., all items in the set $I_1 \cup I_2$.

| Case | Description |
|------|-------------|
| 1 | Both items $(l, m)$ in $I_1 \cup I_2$. Branches into the following three sub-cases based on the presence of items in $I_1 \cap I_2$:
1-a: Both items in $I_1 \cap I_2$. The concerned term is $s_1$.
1-b: One item in $I_1 \cap I_2$. Subdivided into 1-b-i (other in $I_1 \setminus I_2$) and 1-b-ii (other in $I_2 \setminus I_1$); concerned terms are $s_2$ and $s_3$.
1-c: No item in $I_1 \cap I_2$. Addresses cases where $l$ and $m$ are in different sets ($I_1 \setminus I_2$ and $I_2 \setminus I_1$); concerned term is $s_4$. |
| 2 | One item in $I_1 \cup I_2$. I.e., either $l$ is $I_1 \cup I_2$ or $m$ is in $I_1 \cup I_2$, leading to sub-cases 2-a and 2-b; concerned term is $s_5$. |
| 3 | No item in $I_1 \cup I_2$. Addresses the scenario where neither $l$ nor $m$ is in $I_1 \cup I_2$; value trivially zero. |

**Table 4:** Case categorization for the proof of Algorithms 2 and 3 based on item pair ranks, where $I_1$ and $I_2$ are the sets of items for top-k rankings $\pi_1$ and $\pi_2$, respectively.

---

**Definition 1.** *Algorithm 2 and 3 and utilize following notations.*

- $I_1$ and $I_2$ are the sets of items in rankings $\pi_1$ and $\pi_2$, respectively.

- $\sigma_1 \in \Pi^{|I_1|}$ and $\tau_1 \in \Pi^{|I_1 \cap I_2|}$ are the full rankings of $I_1$ and $I_1 \cap I_2$, both consistent with the input top-k ranking $\pi_1$. I.e., relative ranks of items is same yielding $\forall l, m \in I_1 \cap I_2$, $\mathfrak{p}_{i>j}^{\pi_1} = \mathfrak{p}_{i>j}^{\tau_1}$.

- Analogously, $\sigma_2$ and $\tau_2$ are constructed utilizing the set $I_2$ and ranking $\pi_2$.

---

**Algorithm 2** Computing Weighted Convolutional Kendall Kernel

**Input:** Two permutations $\pi_1, \pi_2 \in \Pi^k$. Ranking weighting function $w_s : [n] \times [n] \mapsto \mathbb{R}^n$ and its one dimensional marginals are $w_s(\ell, \cdot) = \frac{1}{n-k} \sum_{j=k+1}^{n} w_s(\ell, j)$.

**Output:** Convolutional Weighted Kendall kernel $k^{wck}(\pi_1, \pi_2)$.

– Let $I_1$ and $I_2$ be the sets of items in rankings $\pi_1$ and $\pi_2$, respectively.

1: **if** $|I_1 \cap I_2| \geq 2$ **then**
2: $\quad s_1(\pi_1, \pi_2) = \frac{1}{\binom{n}{2}} \sum_{1 \leq l < m \leq n | l, m \in I_1 \cap I_2} w_s(\mathbf{p}_l^{\pi_1}, \mathbf{p}_m^{\pi_1}) \cdot w_s(\mathbf{p}_l^{\pi_2}, \mathbf{p}_m^{\pi_2}) \cdot \eta_{l,m}(\pi_1, \pi_2)$
3: **end if**
4: **if** $|I_1 \cap I_2| \geq 1$ and $|I_1 \setminus I_2| \geq 1$ **then**
5: $\quad s_2(\pi_1, \pi_2) = \frac{1}{\binom{n}{2}} \cdot \sum_{l \in I_1 \cap I_2 | m \in I_1 \setminus I_2} w_s(\mathbf{p}_l^{\pi_1}, \mathbf{p}_m^{\pi_1}) \cdot w_s(\mathbf{p}_l^{\pi_2}, \cdot) \left( \mathfrak{p}_{l<m}^{\pi_1} - \mathfrak{p}_{l>m}^{\pi_1} \right)$
6: **end if**
7: **if** $|I_1 \cap I_2| \geq 1$ and $|I_2 \setminus I_1| \geq 1$ **then**
8: $\quad s_3(\pi_1, \pi_2) = \frac{1}{\binom{n}{2}} \cdot \sum_{l \in I_1 \cap I_2 | m \in I_2 \setminus I_1} w_s(\mathbf{p}_l^{\pi_1}, \cdot) \cdot w_s(\mathbf{p}_l^{\pi_2}, \mathbf{p}_m^{\pi_2}) \cdot \left( \mathfrak{p}_{l<m}^{\pi_2} - \mathfrak{p}_{l>m}^{\pi_2} \right)$
9: **end if**
10: **if** $|I_1 \setminus I_2| \geq 1$ and $|I_2 \setminus I_1| \geq 1$ **then**
11: $\quad s_4(\pi_1, \pi_2) = -\frac{1}{\binom{n}{2}} \cdot \sum_{l \in I_1 \setminus I_2 | m \in I_2 \setminus I_1} w_s(\mathbf{p}_l^{\pi_1}, \cdot) \cdot w_s(\mathbf{p}_m^{\pi_2}, \cdot)$
12: **end if**
13: **if** $|I_1 \cap I_2| \geq 1$ and $|[n] \setminus (I_1 \cup I_2)| \geq 1$ **then**
14: $\quad s_5(\pi_1, \pi_2) = \frac{1}{\binom{n}{2}} \cdot (n - |I_1 \cup I_2|) \cdot \sum_{l \in I_1 \cap I_2} w_s(\mathbf{p}_l^{\pi_1}, \cdot) \cdot w_s(\mathbf{p}_l^{\pi_2}, \cdot)$
15: **end if**
16: $k^{wck}(\pi_1, \pi_2) = s_1(\pi_1, \pi_2) + s_2(\pi_1, \pi_2) + s_3(\pi_1, \pi_2) + s_4(\pi_1, \pi_2) + s_5(\pi_1, \pi_2)$

---

**Case 1:** The pair $(l, m) \in I_1 \cup I_2$ falls within the top-k, leading to three distinct cases. Below, we provide $s_i$ terms for each case as given in Table 4.

**Case 1-a:** Two items in $I_1 \cap I_2$, meaning both $l$ and $m$ belong to $I_1 \cap I_2$. Using Claim 3 regarding the feature vector $\phi^{wck}$, we simplify $s_1$ as follows:

$$s_1(\pi_1, \pi_2) = \sum_{1 \leq l < m \leq n | l, m \in I_1 \cap I_2} \phi_{l,m}^{wck}(\pi_1) \cdot \phi_{l,m}^{wck}(\pi_2)$$

$$= \sum_{1 \leq l < m \leq n | l, m \in I_1 \cap I_2} \frac{1}{\sqrt{\binom{n}{2}}} \cdot w_s(\mathbf{p}_l^{\pi_1}, \mathbf{p}_m^{\pi_1}) \cdot \left(\mathfrak{p}_{l<m}^{\pi_1} - \mathfrak{p}_{l>m}^{\pi_1}\right)$$

$$\cdot \frac{1}{\sqrt{\binom{n}{2}}} \cdot w_s(\mathbf{p}_l^{\pi_2}, \mathbf{p}_m^{\pi_2}) \cdot \left(\mathfrak{p}_{l<m}^{\pi_2} - \mathfrak{p}_{l>m}^{\pi_2}\right)$$

$$= \frac{1}{\binom{n}{2}} \sum_{1 \leq l < m \leq n | l, m \in I_1 \cap I_2} w_s(\mathbf{p}_l^{\pi_1}, \mathbf{p}_m^{\pi_1}) \cdot w_s(\mathbf{p}_l^{\pi_2}, \mathbf{p}_m^{\pi_2}) \cdot \eta_{l,m}(\pi_1, \pi_2). \quad (12)$$

**Case 1-b:** When one item is in $I_1 \cap I_2$, the other must reside either in $I_1 \setminus I_2$ or $I_2 \setminus I_1$, thus leading to two distinct sub-cases. This is specified in Table 4. Concretely, if the other item is in $I_1 \setminus I_2$, it contributes to the $s_2$ terms, whereas if it's in $I_2 \setminus I_1$, it contributes to the $s_3$ terms.

Corresponding to Case 1-b-i, when the other item is in $I_1 \cap I_2$, i.e., $s_2$ is the term corresponding to indices where $l$ is in $I_1 \cap I_2$ and $m$ in $I_1 \setminus I_2$, or the reverse, represented by partial sums $u$ and $v$. For the partial sum $u$, with $l$ in $I_1 \cap I_2$ and $m$ in $I_1 \setminus I_2$, we find that $\mathbf{p}_l^{\pi_2}$ is in $[k]$, while $\mathbf{p}_m^{\pi_2}$ is not. The simplification of $u$ proceeds using Claim 3 as follows:

$$u = \sum_{1 \leq l < m \leq n | l \in I_1 \cap I_2 | m \in I_1 \setminus I_2} \frac{1}{\sqrt{\binom{n}{2}}} \cdot w_s(\mathbf{p}_l^{\pi_1}, \mathbf{p}_m^{\pi_1}) \cdot \left(\mathfrak{p}_{l<m}^{\pi_1} - \mathfrak{p}_{l>m}^{\pi_1}\right)$$

$$\cdot \frac{1}{\sqrt{\binom{n}{2}}} \cdot w_s(\mathbf{p}_l^{\pi_2}, \cdot) \left(\mathfrak{p}_{l<m}^{\pi_2} - \mathfrak{p}_{l>m}^{\pi_2}\right)$$

$$= \frac{1}{\binom{n}{2}} \sum_{1 \leq l < m \leq n | l \in I_1 \cap I_2 | m \in I_1 \setminus I_2} w_s(\mathbf{p}_l^{\pi_1}, \mathbf{p}_m^{\pi_1}) \cdot w_s(\mathbf{p}_l^{\pi_2}, \cdot) \left(\mathfrak{p}_{l<m}^{\pi_1} - \mathfrak{p}_{l>m}^{\pi_1}\right).$$

Similarly, the partial sum $v$ can be simplified as follows:

$$v = \sum_{1 \leq l < m \leq n | m \in I_1 \cap I_2 | l \in I_1 \setminus I_2} \frac{1}{\sqrt{\binom{n}{2}}} \cdot w_s(\mathbf{p}_l^{\pi_1}, \mathbf{p}_m^{\pi_1}) \cdot \left(\mathfrak{p}_{l<m}^{\pi_1} - \mathfrak{p}_{l>m}^{\pi_1}\right)$$

$$\cdot \frac{-1}{\sqrt{\binom{n}{2}}} \cdot w_s(\mathbf{p}_l^{\pi_2}, \cdot) \cdot \left(\mathfrak{p}_{l<m}^{\pi_2} - \mathfrak{p}_{l>m}^{\pi_2}\right)$$

$$= \frac{-1}{\binom{n}{2}} \sum_{1 \leq l < m \leq n | m \in I_1 \cap I_2 | l \in I_1 \setminus I_2} w_s(\mathbf{p}_l^{\pi_1}, \mathbf{p}_m^{\pi_1}) \cdot w_s(\mathbf{p}_l^{\pi_2}, \cdot) \cdot \left(\mathfrak{p}_{l<m}^{\pi_1} - \mathfrak{p}_{l>m}^{\pi_1}\right)$$

$$= \frac{-1}{\binom{n}{2}} \sum_{1 \leq m < l \leq n | l \in I_1 \cap I_2 | m \in I_1 \setminus I_2} w_s(\mathbf{p}_m^{\pi_1}, \mathbf{p}_l^{\pi_1}) \cdot w_s(\mathbf{p}_m^{\pi_2}, \cdot) \cdot \left(\mathfrak{p}_{m<l}^{\pi_1} - \mathfrak{p}_{m>l}^{\pi_1}\right)$$

$$= \frac{1}{\binom{n}{2}} \sum_{1 \leq m < l \leq n | l \in I_1 \cap I_2 | m \in I_1 \setminus I_2} w_s(\mathbf{p}_l^{\pi_1}, \mathbf{p}_m^{\pi_1}) \cdot w_s(\mathbf{p}_l^{\pi_2}, \cdot) \left(\mathfrak{p}_{l<m}^{\pi_1} - \mathfrak{p}_{l>m}^{\pi_1}\right).$$

In the above, the first two lines use results from Claim 3 and use similarity of $w_s$. In the following line, $l$ and $m$ are exchanged. Lastly, the negative sign is pushed into the indicator functions to make the summand function of this partial sum $v$ similar to the partial sum $u$, and the similarity of the $w_s$ is utilized. The above partial sums simplify $s_2$ as follows:

$$s_2(\pi_1, \pi_2) = \frac{1}{\binom{n}{2}} \cdot \sum_{l \in I_1 \cap I_2 | m \in I_1 \setminus I_2} w_s(\mathbf{p}_l^{\pi_1}, \mathbf{p}_m^{\pi_1}) \cdot w_s(\mathbf{p}_l^{\pi_2}, \cdot) \left(\mathfrak{p}_{l<m}^{\pi_1} - \mathfrak{p}_{l>m}^{\pi_1}\right). \quad (13)$$

Analogously, in Case 1-b-ii, we deduce the corresponding term $s_3$ for the pair of indices as described in Table 4 through symmetry. Specifically, the term $s_3$ can be outlined as follows:

$$s_3(\pi_1, \pi_2) = \frac{1}{\binom{n}{2}} \cdot \sum_{l \in I_1 \cap I_2 | m \in I_2 \setminus I_1} w_s(\mathbf{p}_l^{\pi_1}, \cdot) \cdot w_s(\mathbf{p}_l^{\pi_2}, \mathbf{p}_m^{\pi_2}) \cdot \left( \mathfrak{p}_{l<m}^{\pi_2} - \mathfrak{p}_{l>m}^{\pi_2} \right). \tag{14}$$

**Case 1-c:** Both items are outside $I_1 \cap I_2$, specifically, $l \in I_1 \setminus I_2$ and $m \in I_2 \setminus I_1$ or the reverse. Like Case 1-b-i, we divide $s_4$ into partial summations $u$ and $v$. Now, we calculate $u$ under the condition that $l \in I_1 \setminus I_2$ and $m \in I_2 \setminus I_1$.

$$
\begin{aligned}
u &= \sum_{1 \le l < m \le n | l \in I_1 \setminus I_2 | m \in I_2 \setminus I_1} \frac{1}{\sqrt{\binom{n}{2}}} \cdot w_s(\mathbf{p}_l^{\pi_1}, \cdot) \cdot \left( \mathfrak{p}_{l<m}^{\pi_1} - \mathfrak{p}_{l>m}^{\pi_1} \right) \\
&\qquad\qquad \cdot \frac{1}{\sqrt{\binom{n}{2}}} \cdot w_s(\mathbf{p}_m^{\pi_2}, \cdot) \cdot \left( \mathfrak{p}_{l<m}^{\pi_2} - \mathfrak{p}_{l>m}^{\pi_2} \right), \\
&= \frac{1}{\binom{n}{2}} \cdot \sum_{1 \le l < m \le n | l \in I_1 \setminus I_2 | m \in I_2 \setminus I_1} w_s(\mathbf{p}_l^{\pi_1}, \cdot) \cdot (1 - 0) \cdot w_s(\mathbf{p}_m^{\pi_2}, \cdot) \cdot (0 - 1), \\
&= \frac{-1}{\binom{n}{2}} \cdot \sum_{1 \le l < m \le n | l \in I_1 \setminus I_2 | m \in I_2 \setminus I_1} w_s(\mathbf{p}_l^{\pi_1}, \cdot) \cdot w_s(\mathbf{p}_m^{\pi_2}, \cdot).
\end{aligned}
$$

Similarly, we can estimate partial sum $v$ for the set $l \in I_2 \setminus I_1$ & $m \in I_1 \setminus I_2$. Using calculations similar to Case-1-b-i for summing $u$ and $v$, we have:

$$s_4(\pi_1, \pi_2) = \frac{-1}{\binom{n}{2}} \cdot \sum_{l \in I_1 \setminus I_2 | m \in I_2 \setminus I_1} w_s(\mathbf{p}_l^{\pi_1}, \cdot) \cdot w_s(\mathbf{p}_m^{\pi_2}, \cdot). \tag{15}$$

**Case 2:** One item exists in $I_1 \cap I_2$, the other in $[n] \setminus (I_1 \cap I_2)$. It branches into two sub-cases: Case 2-a with one item in $I_1 \cup I_2$, and Case 2-b, where one item outside $I_1 \cap I_2$ but is in $I_1 \cup I_2$. Focusing on Case 2-a, represented by $s_5$, we simplify as follows. This involves two index scenarios, either $l \in I_1 \cap I_2$ and $m \notin I_1 \cup I_2$ or vice versa, represented by partial sums $u$ and $v$. We now simplify $u$ below:

$$
\begin{aligned}
u &= \frac{1}{\binom{n}{2}} \sum_{1 \le l < m \le n | l \in I_1 \cap I_2 | m \notin I_1 \cup I_2} \frac{1}{\sqrt{\binom{n}{2}}} \cdot w_s(\mathbf{p}_l^{\pi_1}, \cdot) \cdot \left( \mathfrak{p}_{l<m}^{\pi_1} - \mathfrak{p}_{l>m}^{\pi_1} \right) \\
&\qquad\qquad \cdot \frac{1}{\sqrt{\binom{n}{2}}} \cdot w_s(\mathbf{p}_l^{\pi_2}, \cdot) \cdot \left( \mathfrak{p}_{l<m}^{\pi_2} - \mathfrak{p}_{l>m}^{\pi_2} \right), \\
&= \frac{1}{\binom{n}{2}} \sum_{1 \le l < m \le n | l \in I_1 \cap I_2 | m \notin I_1 \cup I_2} w_s(\mathbf{p}_l^{\pi_1}, \cdot) \cdot w_s(\mathbf{p}_l^{\pi_2}, \cdot), \\
&= \frac{1}{\binom{n}{2}} \sum_{1 \le l < m \le n | l \in I_1 \cap I_2} w_s(\mathbf{p}_l^{\pi_1}, \cdot) \cdot w_s(\mathbf{p}_l^{\pi_2}, \cdot) \cdot (n - |I_1 \cup I_2|).
\end{aligned}
$$

Using steps similar to the previous case, we get the following value for $s_5$:

$$s_5(\pi_1, \pi_2) = \frac{1}{\binom{n}{2}} \cdot (n - |I_1 \cup I_2|) \cdot \sum_{l \in I_1 \cap I_2} w_s(\mathbf{p}_l^{\pi_1}, \cdot) \cdot w_s(\mathbf{p}_l^{\pi_2}, \cdot). \tag{16}$$

For Case 2-b, $l$ or $m$ are absent from $I_1$ or $I_2$, leading to two sub-scenarios. Consequently, either $\phi_{l,m}^{wck}(\pi_1)$ is zero or $\phi_{l,m}^{wck}(\pi_2)$ is zero. Therefore, these terms don't contribute to the overall WCK kernel value.

**Case 3:** No item is in the top-k, i.e., both $l, m \notin I_1 \cup I_2$. As both items are absent from the top-k in either ranking, the value trivially reduces to zero.

After covering all configurations of $l$ and $m$, we incorporate results from Equations 12, 13, 14, 15, and 16. This integration yields the expression $k^{wck}(\pi_1, \pi_2) = \sum_{i=1}^{5} s_i(\pi_1, \pi_2)$, where each term $s_i$ matches precisely with its corresponding expression in Algorithm 2. The proof for the correctness of Algorithm 2 is complete, as each term $s_i$ corresponds to its respective expression in the algorithm. Regarding the time complexity of Algorithm 2, each term $s_i$ sums at most $k^2$ quantities, and each quantity summed can be computed in $\mathcal{O}(1)$ time. Therefore, the computation time required for Algorithm 2 is $\mathcal{O}(k^2)$. $\qquad\square$

### A.2.2 Efficiently Computing the Convolutional Kendall Kernel

This section provides Algorithm 3 for computing the convolutional Kendall kernel, as specified in Equation 3. Later, its efficiency and accuracy are proved in Claim 4.

---

**Algorithm 3** Computing Convolutional Kendall Kernel [10]

---

**Input:** Two top-k rankings $\pi_1, \pi_2 \in \Pi^k$.
**Output:** Convolutional Kendall kernel $k^{ck}(\pi_1, \pi_2)$.
  − Let $I_1$ and $I_2$ be the sets of items in rankings $\pi_1$ and $\pi_2$, respectively.
  − Let $\sigma_1 \in \Pi^{|I_1|}$ and $\tau_1 \in \Pi^{|I_1 \cap I_2|}$ be the full rankings of $I_1$ and $I_1 \cap I_2$, both consistent with the input top-k ranking $\pi_1$.
  − Analogously, construct $\sigma_2$ and $\tau_2$ utilizing the set $I_2$ and ranking $\pi_2$.
1: **if** $|I_1 \cap I_2| \geq 2$ **then**
2:     $s_1(\pi_1, \pi_2) = \frac{1}{\binom{n}{2}} \cdot \binom{|I_1 \cap I_2|}{2} \cdot k^{sk}(\tau_1, \tau_2)$
3: **end if**
4: **if** $|I_1 \cap I_2| \geq 1$ and $|I_1 \setminus I_2| \geq 1$ **then**
5:     $s_2(\pi_1, \pi_2) = \frac{1}{\binom{n}{2}} \cdot \sum_{l \in I_1 \cap I_2} 2 \cdot (\sigma_1(l) - \tau_1(l)) - k + |I_1 \cap I_2|$
6: **end if**
7: **if** $|I_1 \cap I_2| \geq 1$ and $|I_2 \setminus I_1| \geq 1$ **then**
8:     $s_3(\pi_1, \pi_2) = \frac{1}{\binom{n}{2}} \cdot \sum_{l \in I_1 \cap I_2} 2 \cdot (\sigma_2(l) - \tau_2(l)) - k + |I_1 \cap I_2|$
9: **end if**
10: $s_4(\pi_1, \pi_2) = -\frac{1}{\binom{n}{2}} \cdot |I_1 \setminus I_2| \cdot |I_1 \setminus I_2|$
11: $s_5(\pi_1, \pi_2) = \frac{1}{\binom{n}{2}} \cdot |I_1 \cap I_2| \cdot |[n] \setminus (I_1 \cup I_2)|$
12: $k^{ck}(\pi_1, \pi_2) = s_1(\pi_1, \pi_2) + s_2(\pi_1, \pi_2) + s_3(\pi_1, \pi_2) + s_4(\pi_1, \pi_2) + s_5(\pi_1, \pi_2)$

---

**Claim 4.** *Algorithm 3 computes the convolutional Kendall kernel (as given in the Equation 3) with a computational complexity of $\mathcal{O}(k^2)$.*

*Proof.* To establish the correctness of Algorithm 3, we will adopt the same proof approach as the one used for Claim 1 concerning Algorithm 2. Specifically, we will adhere to the earlier categorization in Table 4 and notations given in Definition 1. Since the CK kernel can be derived by uniformly setting the weight function $w_s(i, j) = 1$, we will insert them in $s_i$ terms as given in Algorithm 2. These cases will be revisited and simplified by applying the condition $w_s(i, j) = 1$. Note that this also implies its one-direction marginal weights to be 1, i.e., $w_s(i, \cdot) = 1$

**Simplifying the $s_1$ Term:** For the WCK kernel, Case 1-a leads to the expression of $s_1$ as stated in Equation 12. In this case, when two items, specifically $l$ and $m$, are both in the intersection $I_1 \cap I_2$, it implies that $\mathbf{p}_l^{\pi_1}$, $\mathbf{p}_m^{\pi_1}$, $\mathbf{p}_l^{\pi_2}$, and $\mathbf{p}_m^{\pi_2}$ all rank within the top-k, denoted as $[k]$. We simplify the $s_1$ term for CK kernel as follows:

$$s_1(\pi_1, \pi_2) = \frac{1}{\binom{n}{2}} \sum_{1 \le l < m \le n | l, m \in I_1 \cap I_2} w_s(\mathbf{p}_i^{\pi_l}, \mathbf{p}_m^{\pi_1}) \cdot w_s(\mathbf{p}_l^{\pi_2}, \mathbf{p}_m^{\pi_2}) \cdot \eta_{l,m}(\pi_1, \pi_2)$$

$$= \frac{1}{\binom{n}{2}} \sum_{1 \le l < m \le n | l, m \in I_1 \cap I_2} \eta_{l,m}(\pi_1, \pi_2)$$

$$= \frac{1}{\binom{n}{2}} \sum_{1 < l' < m' \le |I_1 \cap I_2|} \eta_{l',m'}(\tau_1, \tau_2) = \frac{\binom{|I_1 \cap I_2|}{2}}{\binom{n}{2}} k^{sk}(\tau_1, \tau_2). \tag{17}$$

The simplification process begins by assigning unit rank weights in the first line, i.e., $\mathbf{w}_i = 1$. Following this, by relabeling the items in $I_1 \cap I_2$ and using $\tau_1$ and $\tau_2$, which are the rankings of $\pi_1$ and $\pi_2$ limited to the set $I_1 \cap I_2$ as defined in Definition 1, it is established that $\eta_{l',m'}(\tau_1, \tau_2) = \eta_{l,m}(\pi_1, \pi_2)$. This is because the relative order of any pair of items is maintained in $\tau_1$ and $\tau_2$. Consequently, this leads to the final simplification to a scaled value of the standard Kendall kernel $k^{sk}$, as given in Equation 1.

**Simplifying the $s_2$ and $s_3$ Terms:** The $s_2$ and $s_3$ terms are obtained for Case 1-b, which is for case when one item is in $I_1 \cap I_2$ and the other item is either in $I_1 \setminus I_2$ or $I_2 \setminus I_1$. We divide this into two sub-cases. Case 1-b-i: The other item is in $I_1 \setminus I_2$, with $s_2$ representing the summation terms derived from the CK's inner product. Case 1-b-ii: The other item is $I_2 \setminus I_1$, where $s_3$ denotes the summation terms. We simplify the $s_2$ term for the CK kernel as follows:

$$s_2(\pi_1, \pi_2) = \frac{1}{\binom{n}{2}} \sum_{l \in I_1 \cap I_2 | m \in I_1 \setminus I_2} w_s(\mathbf{p}_l^{\pi_1}, \mathbf{p}_m^{\pi_1}) \cdot w_s(\mathbf{p}_m^{\pi_2}, \cdot) \left( \mathfrak{p}_{l<m}^{\pi_1} - \mathfrak{p}_{l>m}^{\pi_1} \right)$$

$$= \frac{1}{\binom{n}{2}} \sum_{l \in I_1 \cap I_2 | m \in I_1 \setminus I_2} \left( \mathfrak{p}_{l<m}^{\pi_1} - \mathfrak{p}_{l>m}^{\pi_1} \right)$$

$$= \underbrace{\frac{1}{\binom{n}{2}} \sum_{l \in I_1 \cap I_2 | m \in I_1 \setminus I_2} \mathfrak{p}_{l<m}^{\pi_1}}_{:=u} - \underbrace{\frac{1}{\binom{n}{2}} \sum_{l \in I_1 \cap I_2 | m \in I_1 \setminus I_2} \mathfrak{p}_{l>m}^{\pi_1}}_{:=v}.$$

Next, we examine the terms $u$ and $v$ in detail, starting with $u$. The term $u$, which corresponds to $\mathfrak{p}_{l<m}^{\pi_1}$, signifies instances where item $l$ is ranked before item $m$ in the top-k ranking $\pi_1$. This can be derived from the observation that $\sigma_1(l) - 1$ items are positioned before item $l$ in the set $I_1$. Out of these items, $\tau_1(l) - 1$ also belong to the intersection $I_1 \cap I_2$. This follows from the definition of the full rankings $\sigma_1$ and $\tau_1$ on the set $I_1$ and the intersection $I_1 \cap I_2$, respectively. Consequently, it can be concluded that $\sigma_1(l) - \tau_1(l)$ items from the set difference $I_1 \setminus I_2$ are ranked before item $l$. The second term, $v$, corresponds to $\mathfrak{p}_{l>m}^{\pi_1}$ and involves a calculation that takes into account the items ranked after the $l$-th item in the set $I$. Specifically, there are $k - \sigma_1(l)$ items following the $l$-th item. Within the intersection $I_1 \cap I_2$, the number of items before $l$ is given by $|I_1 \cap I_2| - \tau_1(l)$. Therefore, the expression $(k - \sigma_1(l)) - (|I_1 \cap I_2| - \tau_1(l))$ represents the count of elements that are positioned after $l$ in the set difference $I_1 \setminus I_2$.

Combining the above calculations for both terms $u$ and $v$, the $s_2$ term for the CK kernel can be simplified as follows:

$$s_2(\pi_1, \pi_2) = \frac{1}{\binom{n}{2}} \sum_{l \in I_1 \cap I_2} 2 \cdot (\sigma_1(l) - \tau_1(l)) - k + |I_1 \cap I_2|. \tag{18}$$

Using the symmetry between Case 1-b-i and Case 1-b-ii, we can simplify $s_3$ for the CK kernel as follows:

$$s_3(\pi_1, \pi_2) = \frac{1}{\binom{n}{2}} \sum_{l \in I_1 \cap I_2} 2 \cdot (\sigma_2(l) - \tau_2(l)) - k + |I_1 \cap I_2|. \tag{19}$$

**Simplifying the $s_4$ and $s_5$ Terms:** We simplify the $s_4$ and $s_5$ terms for the CK kernel starting from Equation 15 and Equation 16, respectively, as follows:

$$s_4(\pi_1, \pi_2) = \frac{-1}{\binom{n}{2}} \cdot \sum_{l \in I_1 \setminus I_2 | m \in I_2 \setminus I_1} w_s(\mathbf{p}_l^{\pi_1}, \cdot) \cdot w_s(\mathbf{p}_m^{\pi_2}, \cdot) = \frac{-|I_1 \setminus I_2| \cdot |I_2 \setminus I_1|}{\binom{n}{2}} \tag{20}$$

$$s_5(\pi_1, \pi_2) = \frac{1}{\binom{n}{2}} \cdot (n - |I_1 \cup I_2|) \cdot \sum_{l \in I_1 \cap I_2} w_s(\mathbf{p}_l^{\pi_1}, \cdot) \cdot w_s(\mathbf{p}_l^{\pi_2}, \cdot) = \frac{|I_1 \cap I_2| \cdot |[n] \setminus (I_1 \cup I_2)|}{\binom{n}{2}}. \tag{21}$$

We have obtained the values of all the simplified $s_i$ terms for the CK kernel in Equations 17, 18, 19, 20, and 21. By combining these terms, we get $k^{ck}(\pi_1, \pi_2) = \sum_{i=1}^{5} s_i(\pi_1, \pi_2)$, where each term $s_i$ precisely matches its corresponding expression in Algorithm 3. This completes the proof of the correctness of Algorithm 3. Regarding its time complexity, each term $s_i$ sums at most $k^2$ quantities, and each quantity can be computed in $\mathcal{O}(1)$ time. Therefore, the time required for Algorithm 3 to compute the CK kernel is $\mathcal{O}(k^2)$. $\square$

### A.3 Fast Matrix-Vector Multiplication with Kendall Kernel Matrix on Top-k Rankings

This section revisits Theorem 1 about the fact matrix-vector multiplication time for the Kendall kernel matrix for top-k rankings. Specifically, we aim to eliminate the $\mathbf{mvm}(K_X)$'s dependence on the number of items, i.e., $n$ on and linear dependence in the number of rounds, i.e., $T$, as claimed in Theorem 1.

> **Theorem 1.** *For the WCK kernel with product-symmetric weights $w_{ps}$, the computational complexity of multiplying the kernel matrix $K_{X_t}$ with any admissible vector is $\mathcal{O}(k^2 t)$, i.e., $\mathbf{mvm}(K_{X_t}) = \mathcal{O}(k^2 t)$, where $X_t$ is any arbitrary set of $t$ top-k rankings.*

*Proof.* The cornerstone of this proof lies in the computation of the WCK kernel, as delineated in Algorithm 2. This algorithm requires only $\mathcal{O}(k^2)$ computation. For brevity, we write $X$ to represent $X_T$, and the proof follows for any $X_t$, i.e., any value of $t$, not just $T$.

As also suggested previously, we will demonstrate through the equation $K_X = (\Phi_X^a)^T \Phi_X^b$, where both matrices $\Phi_X^a$ and $\Phi_X^b$ have columns with only $\mathcal{O}(k^2)$ non-zero entries. Consequently, this leads to the computational complexity of matrix-vector multiplication, denoted as $\mathbf{mvm}(K_X)$, being $\mathcal{O}(k^2 \cdot T)$.

From Algorithm 2, we know that each entry of the kernel matrix $k(\pi_1, \pi_2)$, can be expressed as a sum $\sum_{i=1}^{5} s_i(\pi_1, \pi_2)$. Assuming each $s_i(\pi_1, \pi_2)$ equals $\phi^{a_i}(\pi_1)^T \phi^{b_i}(\pi_2)$, and considering that all vectors $\phi^{a_i}$ and $\phi^{b_i}$ exhibit this property, we can express $K_X$ as $(\Phi_X^a)^T \Phi_X^b$. Here, the $i^{th}$ row of $(\Phi_X^a)^T$ and the $j^{th}$ column of $\Phi_X^b$ are represented by $[\phi^{a_1}(\pi_i)^T, \cdots, \phi^{a_5}(\pi_i)^T]$ and $[\phi^{b_1}(\pi_j), \cdots, \phi^{b_5}(\pi_j)]$, respectively. Therefore, the overall mvm complexity can be characterized by the sparsity of the vectors $\phi^{a_i}$ and $\phi^{b_i}$, as is formalized in the claim presented below.

> **Claim 5.** *Consider a kernel matrix $K_X$ corresponding to any set $X$ of cardinality $T$. Each entry of $K_X$, denoted as $k(x_1, x_2)$, is defined by the sum $\sum_{i=1}^{5} s_i(x_1, x_2)$, where each $s_i(x_1, x_2)$ is the result of the dot product $\phi^{a_i}(x_1)^T \phi^{b_i}(x_2)$, where, $\phi^{a_i}$ and $\phi^{b_i}$ are vectors characterized by having $\mathcal{O}(z)$ non-zero entries. Given this structure, the matrix-vector multiplication complexity for $K_X$ is $O(nnz \cdot T)$, i.e., $\mathbf{mvm}(K_X) = O(z \cdot T)$.*

*Proof.* We will demonstrate this in the following discussion by concentrating on the $k^{\text{th}}$ entry of the output vector, specifically $K_X \mathbf{v}$, for any arbitrary vector $\mathbf{v}$:

$$(K_X \mathbf{v})_k = \sum_j K_X(k, j) v_j = \sum_j \left( \sum_{i=1}^{5} s_i(\pi_k, \pi_j) \right) v_j = \sum_j \left( \sum_{i=1}^{5} \phi^{a_i}(\pi_k)^T \phi^{b_i}(\pi_j) \right) v_j,$$

$$= \sum_{i=1}^{5} \left( \sum_{j} \phi^{a_i}(\pi_k)^T \phi^{b_i}(\pi_j) v_j \right) = \sum_{i=1}^{5} \phi^{a_i}(\pi_k)^T \left( \sum_{j} \phi^{b_i}(\pi_j) v_j \right).$$

Given that for all $i$, $\phi^{b_i}$ possesses only $\mathcal{O}(z)$ non-zero entries for any $\pi_j$, the computation of $\sum_j \phi^{b_i}(\pi_j) v_j$ requires $\mathcal{O}(z)$ operations. This implies that the expression $\sum_j \phi^{b_i}(\pi_j) v_j$ also necessitates $\mathcal{O}(z)$ computation. Applying a similar rationale to $\phi^{a_i}$, it follows that computing $(K_X v)_k$ demands only $\mathcal{O}(z)$ operations. Extending this argument to all entries of the output vector, it is evident that computing $K_X \mathbf{v}$ requires only $\mathcal{O}(z \cdot T)$ computation. $\qquad \square$

Utilizing Claim 5, it suffices to complete the proof by showcasing that these exist vectors $\phi^{a_i}$ and $\phi^{b_i}$, each with only $\mathcal{O}(k^2)$ non-zero elements, corresponding to each $s_i$ as specified in Algorithm 2. Additionally, these vectors ensure that $s_i(\pi_1, \pi_2) = \phi^{a_i}(\pi_1)^T \phi^{b_i}(\pi_2)$. We will next establish such vectors for all $s_i$ terms. Starting with the $s_1$ term below.

**Showcasing** $s_1(\pi_1, \pi_2) = \phi^{a_1}(\pi_1)^T \phi^{b_1}(\pi_2)$ for sparse $\phi^{a_1}(\pi_1)$ and $\phi^{b_1}(\pi_2)$ vectors. We begin by manipulating $s_1$, as defined in Equation 12. For the sake of brevity, their scalar factors will be omitted in the following explanation.

$$
\begin{aligned}
s_1(\pi_1, \pi_2) =& \sum_{1 \leq l < m \leq n | l, m \in I_1 \cap I_2} w_s(\mathbf{p}_l^{\pi_1}, \mathbf{p}_m^{\pi_1}) \cdot w_s(\mathbf{p}_l^{\pi_2}, \mathbf{p}_m^{\pi_2}) \cdot \eta_{l,m}(\pi_1, \pi_2), \\
=& \sum_{1 \leq l < m \leq n | l, m \in I_1 \cap I_2} w_s(\mathbf{p}_l^{\pi_1}, \mathbf{p}_m^{\pi_1}) \cdot w_s(\mathbf{p}_l^{\pi_2}, \mathbf{p}_m^{\pi_2}) \cdot (\mathfrak{p}_{l<m}^{\pi_1} - \mathfrak{p}_{l>m}^{\pi_1}) \cdot (\mathfrak{p}_{l<m}^{\pi_2} - \mathfrak{p}_{l>m}^{\pi_2}), \\
=& \sum_{1 \leq l < m \leq n | l, m \in I_1 \cap I_2} w_s(\mathbf{p}_l^{\pi_1}, \mathbf{p}_m^{\pi_1}) \cdot (\mathfrak{p}_{i<j}^{\pi_1} - \mathfrak{p}_{l>m}^{\pi_1}) \cdot w_s(\mathbf{p}_l^{\pi_2}, \mathbf{p}_m^{\pi_2}) \cdot (\mathfrak{p}_{l<m}^{\pi_2} - \mathfrak{p}_{l>m}^{\pi_2}), \\
=& \sum_{1 \leq l < m \leq n} \underbrace{w_s(\mathbf{p}_l^{\pi_1}, \mathbf{p}_m^{\pi_1}) \cdot (\mathfrak{p}_{i<j}^{\pi_1} - \mathfrak{p}_{l>m}^{\pi_1}) \cdot \mathbf{1}_{\mathbf{p}_l^{\pi_1}, \mathbf{p}_m^{\pi_1} \in [k]}}_{:=\phi_{l,m}^{a_1}(\pi_1)} \\
& \cdot \underbrace{w_s(\mathbf{p}_l^{\pi_2}, \mathbf{p}_m^{\pi_2}) \cdot (\mathfrak{p}_{l<m}^{\pi_2} - \mathfrak{p}_{l>m}^{\pi_2}) \cdot \mathbf{1}_{\mathbf{p}_l^{\pi_2}, \mathbf{p}_m^{\pi_2} \in [k]}}_{:=\phi_{l,m}^{b_1}(\pi_2)}, \\
=& (\phi^{a_1}(\pi_1)^T \phi^{b_1}(\pi_2)).
\end{aligned}
\tag{22}
$$

Both $\phi^{a_1}$ and $\phi^{b_1}$ are sparse by design, taking non-zero values only when $l$ and $m$ appear in the top-k rankings. This demonstrates the existence of sparse vectors for the $s_1$ term. Next, we will establish the same for the $s_2$ and $s_3$ terms.

**Showcasing sparse vectors for $s_2$ and $s_3$.** We begin by manipulating $s_2$, as defined in Equation 13, while ignoring its scalar factor. We will exploit symmetry between $s_2$ and $s_3$ terms.

$$
\begin{aligned}
& s_2(\pi_1, \pi_2) \\
=& \sum_{l \in I_1 \cap I_2 | m \in I_1 \setminus I_2} w_s(\mathbf{p}_l^{\pi_1}, \mathbf{p}_m^{\pi_1}) \cdot w_s(\mathbf{p}_l^{\pi_2}, \cdot) \left( \mathfrak{p}_{l<m}^{\pi_1} - \mathfrak{p}_{l>m}^{\pi_1} \right), \\
=& \sum_{l \in I_1 \cap I_2} w_s(\mathbf{p}_l^{\pi_2}, \cdot) \sum_{m \in I_1 \setminus I_2} w_s(\mathbf{p}_l^{\pi_1}, \mathbf{p}_m^{\pi_1}) \left( \mathfrak{p}_{l<m}^{\pi_1} - \mathfrak{p}_{l>m}^{\pi_1} \right), \\
=& \sum_{l \in I_1 \cap I_2} w_s(\mathbf{p}_l^{\pi_2}, \cdot) \left( \sum_{m \in I_1} w_s(\mathbf{p}_l^{\pi_1}, \mathbf{p}_m^{\pi_1}) \left( \mathfrak{p}_{l<m}^{\pi_1} - \mathfrak{p}_{l>m}^{\pi_1} \right) - \sum_{m \in I_1 \cap I_2} w_s(\mathbf{p}_l^{\pi_1}, \mathbf{p}_m^{\pi_1}) \left( \mathfrak{p}_{l<m}^{\pi_1} - \mathfrak{p}_{l>m}^{\pi_1} \right) \right), \\
=& \sum_{l \in [n]} \underbrace{\mathbf{1}_{\mathbf{p}_l^{\pi_2} \in [k]} w_s(\mathbf{p}_l^{\pi_2}, \cdot)}_{:=\phi_l^{b_{21}}(\pi_2)} \underbrace{\mathbf{1}_{\mathbf{p}_l^{\pi_1} \in [k]} \sum_{m \in I_1} w_s(\mathbf{p}_l^{\pi_1}, \mathbf{p}_m^{\pi_1}) \left( \mathfrak{p}_{l<m}^{\pi_1} - \mathfrak{p}_{l>m}^{\pi_1} \right)}_{:=\phi_l^{a_{21}}(\pi_1)}
\end{aligned}
$$

$$- \sum_{l,m \in I_1 \cap I_2} w_s(\mathbf{p}_l^{\pi_2}, \cdot) w_s(\mathbf{p}_l^{\pi_1}, \mathbf{p}_m^{\pi_1}) \left( \mathfrak{p}_{l<m}^{\pi_1} - \mathfrak{p}_{l>m}^{\pi_1} \right), \tag{23}$$

$$= \phi^{a_{21}}(\pi_1)^T \phi^{b_{21}}(\pi_2) - \sum_{l,m \in I_1 \cap I_2} w_s(\mathbf{p}_l^{\pi_2}, \cdot) w_s(\mathbf{p}_l^{\pi_1}, \mathbf{p}_m^{\pi_1}) \left( \mathfrak{p}_{l<m}^{\pi_1} - \mathfrak{p}_{l>m}^{\pi_1} \right),$$

$$= \phi^{a_{21}}(\pi_1)^T \phi^{b_{21}}(\pi_2) + \sum_{l,m \in [n]} \underbrace{-w_s(\mathbf{p}_l^{\pi_2}, \cdot) \mathbf{1}_{\mathbf{p}_l^{\pi_2}, \mathbf{p}_m^{\pi_2} \in [k]}}_{:= \phi_{l,m}^{b_{22}}}$$

$$\cdot \underbrace{w_s(\mathbf{p}_l^{\pi_1}, \mathbf{p}_m^{\pi_1}) \left( \mathfrak{p}_{l<m}^{\pi_1} - \mathfrak{p}_{l>m}^{\pi_1} \right) \mathbf{1}_{\mathbf{p}_l^{\pi_1}, \mathbf{p}_m^{\pi_1} \in [k]}}_{:= \phi_{l,m}^{a_{22}}}, \tag{24}$$

$$= \phi^{a_{21}}(\pi_1)^T \phi^{b_{21}}(\pi_2) + \phi^{a_{22}}(\pi_1)^T \phi^{b_{22}}(\pi_2),$$

$$= \underbrace{[\phi^{a_{21}}(\pi_1); \phi^{a_{22}}(\pi_2)]^T}_{:= \phi^{a_2}(\pi_1)^T} \underbrace{[\phi^{a_{21}}((\pi_2)); \phi^{b_{22}}((\pi_2))]}_{:= \phi^{b_2}(\pi_2} = \phi^{a_2}(\pi_1)^T \phi^{b_2}(\pi_2). \tag{25}$$

Equation 25 demonstrates the existence of vectors $\phi^{a_2}$ and $\phi^{b_2}$ for the $s_2$ term. The vectors $\phi^{a_{21}}$ and $\phi^{a_{22}}$, possessing $\mathcal{O}(k)$ and $\mathcal{O}(k^2)$ non-zero entries respectively, are defined in Equations 23 and 24. Consequently, the $\phi^{a_2}$ vector has $\mathcal{O}(k^2)$ non-zero entries. Similarly, it can be shown that $\phi^{b_2}$ contains $\mathcal{O}(k^2)$ non-zero entries, thus fulfilling the proof requirements for proving the $s_2$ term. For the $s_3$ term, we observe a symmetry between $s_2$ and $s_3$, namely $s_3(\pi_1, \pi_2) = s_2(\pi_2, \pi_1)$. This symmetry makes it trivial to satisfy the requirements, as further highlighted by the following equation:

$$s_3(\pi_1, \pi_2) = s_2(\pi_2, \pi_1) = \phi^{a_2}(\pi_2)^T \phi^{b_2}(\pi_1) = \underbrace{\phi^{b_2}(\pi_1)^T}_{:= \phi^{a_3}(\pi_1)} \underbrace{\phi^{a_2}(\pi_2)}_{:= \phi^{b_3}(\pi_2)} = \phi^{a_3}(\pi_1)^T \phi^{b_3}(\pi_2). \tag{26}$$

**Showcasing sparse vectors** $s_4(\pi_1, \pi_2) = \phi^{4a}(\pi_1)^T \phi^{4b}(\pi_2)$**.** We begin by manipulating the $s_4$ term without scalar, as defined in Equation 15.

$$s_4(\pi_1, \pi_2) = - \sum_{l \in I_1 \setminus I_2} w_s(\mathbf{p}_l^{\pi_1}, \cdot) \cdot w_s(\mathbf{p}_m^{\pi_2}, \cdot),$$

$$= - \sum_{l \in I_1 \setminus I_2} w_s(\mathbf{p}_l^{\pi_1}, \cdot) \cdot \left( \sum_{m \in I_2} w_s(\mathbf{p}_m^{\pi_2}, \cdot) - \sum_{m \in I_1 \cap I_2} w_s(\mathbf{p}_m^{\pi_2}, \cdot) \right).$$

Observing that $\overline{w} := \sum_{m \in I_2} w_s(\mathbf{p}_m^{\pi_2}, \cdot)$ represents a constant value that does not depend on $I_2$, we can further simplify the above expression for $s_4$ as follows:

$$s_4(\pi_1, \pi_2) = - \sum_{l \in I_1 \setminus I_2} w_s(\mathbf{p}_l^{\pi_1}, \cdot) \cdot \left( \overline{w} - \sum_{m \in I_1 \cap I_2} w_s(\mathbf{p}_m^{\pi_2}, \cdot) \right),$$

$$= - \left( \overline{w} - \sum_{l \in I_1 \cap I_2} w_s(\mathbf{p}_l^{\pi_1}, \cdot) \right) \cdot \left( \overline{w} - \sum_{m \in I_1 \cap I_2} w_s(\mathbf{p}_m^{\pi_2}, \cdot) \right),$$

$$= -\overline{w}^2 + \overline{w} \left( \sum_{l \in I_1 \cap I_2} w_s(\mathbf{p}_l^{\pi_1}, \cdot) + \sum_{m \in I_1 \cap I_2} + w_s(\mathbf{p}_m^{\pi_2}, \cdot) \right)$$

$$- \sum_{l \in I_1 \cap I_2} w_s(\mathbf{p}_l^{\pi_1}, \cdot) \sum_{m \in I_1 \cap I_2} w_s(\mathbf{p}_m^{\pi_2}, \cdot). \tag{27}$$

Next, to simplify the above equation, we first focus on the second term and have the following:

$$\overline{w} \left( \sum_{l \in I_1 \cap I_2} w_s(\mathbf{p}_l^{\pi_1}, \cdot) + \sum_{m \in I_1 \cap I_2} w_s(\mathbf{p}_m^{\pi_2}, \cdot) \right)$$

$$= \sum_{l\in[n]} \underbrace{\mathbf{1}_{\mathbf{p}_l^{\pi_1}\in[k]} w_s(\mathbf{p}_l^{\pi_1},\cdot)}_{:=\phi_l^{4a_1}(\pi_1)} \underbrace{\mathbf{1}_{\mathbf{p}_l^{\pi_2}\in[k]}\overline{w}}_{:=\phi_l^{4b_1}(\pi_2)} + \sum_{m\in I_1\cap I_2} \overline{w}\cdot w_s(\mathbf{p}_m^{\pi_2},\cdot),$$

$$= \phi^{4a_1}(\pi_1)^T \phi^{4b_1}(\pi_2) + \sum_{m\in[k]} \underbrace{\mathbf{1}_{\mathbf{p}_m^{\pi_1}\in[k]}\overline{w}}_{:=\phi_m^{4a_2}(\pi_1)} \underbrace{w_s(\mathbf{p}_m^{\pi_1},\cdot)}_{:=\phi_m^{4b_2}(\pi_2)}$$

$$= \phi^{4a_1}(\pi_1)^T \phi^{4b_1}(\pi_2) + \phi^{4a_2}(\pi_1)^T \phi^{4b_2}(\pi_2). \tag{28}$$

Next, we simplify the third and last term in the Equation 27 as follows:

$$\sum_{l\in I_1\cap I_2} w_s(\mathbf{p}_l^{\pi_1},\cdot) \sum_{m\in I_1\cap I_2} w_s(\mathbf{p}_m^{\pi_2},\cdot) = \sum_{l\in[n],m\in[n]} \underbrace{w_s(\mathbf{p}_l^{\pi_1},\cdot)\mathbf{1}_{\mathbf{p}_l^{\pi_1},\mathbf{p}_m^{\pi_1}\in[k]}}_{:=\phi_{l,m}^{4a_3}(\pi_1)} \underbrace{w_s(\mathbf{p}_m^{\pi_2},\cdot)\mathbf{1}_{\mathbf{p}_l^{\pi_2},\mathbf{p}_m^{\pi_2}\in[k]}}_{:=\phi_{l,m}^{4b_3}(\pi_2)},$$

$$= \phi^{4a_3}(\pi_1)^T \phi^{4b_3}(\pi_2). \tag{29}$$

Next, combining the results from Equations 27, 28, and 29, we obtain the following:

$$s_4(\pi_1,\pi_2) = \underbrace{[\overline{w},\phi^{4a_1}(\pi_1);\phi^{4a_1}(\pi_1);\phi^{4a_3}(\pi_1)]^T}_{:=\phi^{4a}(\pi_1)^T} \underbrace{[-\overline{w};\phi^{4b_1}(\pi_2);\phi^{4b_2}(\pi_2);-\phi^{4b_3}(\pi_2)]}_{:=\phi^{4b}(\pi_2)}$$

$$= \phi^{4a}(\pi_1)^T \phi^{4b}(\pi_2). \tag{30}$$

Equation 30 showcases both $\phi^{4a}$ and $\phi^{4b}$ has three components with having only $\mathcal{O}(k^2)$ non-zero entries, thus fulfilling the requirements for the $s_4$ term. Next, we focus on the $s_5$ term.

**Showcasing sparse vectors** $s_5(\pi_1,\pi_2) = \phi^{5a}(\pi_1)^T\phi^{5b}(\pi_2)$. We begin by examining the $s_5$ term, excluding its scalar component, as outlined in Equation 16.

$$s_5(\pi_1,\pi_2) = (n-|I_1\cup I_2|)\cdot \sum_{l\in I_1\cap I_2} w_s(\mathbf{p}_l^{\pi_1},\cdot)\cdot w_s(\mathbf{p}_l^{\pi_2},\cdot),$$

$$= (n-(2k-|I_1\cap I_2|))\cdot \sum_{l\in I_1\cap I_2} w_s(\mathbf{p}_l^{\pi_1},\cdot)\cdot w_s(\mathbf{p}_l^{\pi_2},\cdot),$$

$$= (n-2k)\cdot \sum_{l\in I_1\cap I_2} w_s(\mathbf{p}_l^{\pi_1},\cdot)\cdot w_s(\mathbf{p}_l^{\pi_2},\cdot) + |I_1\cap I_2|\cdot \sum_{l\in I_1\cap I_2} w_s(\mathbf{p}_l^{\pi_1},\cdot)\cdot w_s(\mathbf{p}_l^{\pi_2},\cdot),$$

$$= \sum_{l\in I_1\cap I_2} \sqrt{n-2k}\cdot w_s(\mathbf{p}_l^{\pi_1},\cdot)\cdot \sqrt{n-2k}\cdot w_s(\mathbf{p}_l^{\pi_2},\cdot)$$

$$+ \sum_{l\in I_1\cap I_2} w_s(\mathbf{p}_l^{\pi_1},\cdot)\cdot w_s(\mathbf{p}_l^{\pi_2},\cdot)\cdot|I_1\cap I_2|, \tag{31}$$

$$= \sum_{l\in[n]} \underbrace{\sqrt{n-2k}\cdot w_s(\mathbf{p}_l^{\pi_1},\cdot)\cdot\mathbf{1}_{\mathbf{p}_l^{\pi_1}[k]}}_{:=\phi_l^{5a_1}(\pi_1)}\cdot \underbrace{\sqrt{n-2k}\cdot w_s(\mathbf{p}_l^{\pi_2},\cdot)\cdot\mathbf{1}_{\mathbf{p}_l^{\pi_2}[k]}}_{:=\phi_l^{5b_1}(\pi_2)}$$

$$+ \sum_{l\in I_1\cap I_2} w_s(\mathbf{p}_l^{\pi_1},\cdot)\cdot w_s(\mathbf{p}_l^{\pi_2},\cdot)\cdot|I_1\cap I_2|,$$

$$= \phi^{5a_1}(\pi_1)^T \phi^{5b_1}(\pi_2) + \sum_{l\in I_1\cap I_2} w_s(\mathbf{p}_l^{\pi_1},\cdot)\cdot w_s(\mathbf{p}_l^{\pi_2},\cdot)\cdot \sum_{m\in I_1\cap I_2} 1,$$

$$= (\phi^{5a_1}(\pi_1)^T \phi^{5b_1}(\pi_2) + \sum_{l\in I_1\cap I_2,m\in I_1\cap I_2} w_s(\mathbf{p}_l^{\pi_1},\cdot)\cdot w_s(\mathbf{p}_l^{\pi_2},\cdot),$$

$$= \phi^{5a_1}(\pi_1)^T \phi^{5b_1}(\pi_2) + \sum_{l\in[n],m\in[n]} \underbrace{w_s(\mathbf{p}_l^{\pi_1},\cdot)\cdot\mathbf{1}_{\mathbf{p}_l^{\pi_1},\mathbf{p}_m^{\pi_1}\in[k]}}_{:=\phi_{l,m}^{5a_2}(\pi_1)}\cdot \underbrace{w_s(\mathbf{p}_l^{\pi_2},\cdot)\cdot\mathbf{1}_{\mathbf{p}_l^{\pi_2},\mathbf{p}_m^{\pi_2}\in[k]}}_{:=\phi_{l,m}^{5b_2}(\pi_2)}, \tag{32}$$

$$= \phi^{5a_1}(\pi_1)^T \phi^{5b_1}(\pi_2) + \phi^{5a_2}(\pi_1)^T \phi^{5b_2}(\pi_2),$$
$$= \underbrace{[\phi^{5a_1}(\pi_1); \phi^{5a_2}(\pi_1)]^T}_{:=\phi^{5a}(\pi_1)^T} \underbrace{[\phi^{5b_1}(\pi_2) + \phi^{5b_2}(\pi_2)]}_{:=\phi^{5b}(\pi_2)} = \phi^{5a}(\pi_1)^T \phi^{5b}(\pi_2). \tag{33}$$

The equation shows that $s_5(\pi_1, \pi_2) = \phi^{5a}(\pi_1)^T \phi^{5b}(\pi_2)$, where both $\phi^{5a}$ and $\phi^{5b}$ possess components with a maximum number of non-zero entries, as indicated in Equations 31 and 32. This completes the proof requirements for the $s_5$ term.

By combining the results from Equations 22, 25, 26, 30, and 33, we have demonstrated the existence of vectors $\phi^{a_i}$ and $\phi^{b_i}$, each containing only $\mathcal{O}(k^2)$ non-zero elements, and have established that $s_i(\pi_1, \pi_2) = \phi^{a_i}(\pi_1)^T \phi^{b_i}(\pi_2)$ for each $i \in 1, 2, 3, 4, 5$. In conjunction with Claim 5, this completes the proof. $\qquad \square$

# B   Proposed GP-TopK Bandit Algorithm– Omitted Details

This section includes the proofs that were omitted from Section 4, presented in the following order:

- Section B.1 outlines a brief of Gaussian process regression for any domain.
- Section B.2 summarizes the committed details about the local search utilized for optimizing the UCB function.
- Section B.3 provides the removed proof for the Theorem 2 concerning the overall time for the bandit algorithm.
- Section B.4 provides the proof for Theorem 3 concerning regret analysis of the proposed bandit algorithm.

## B.1   Gaussian Process Regression

In GP regression [22], the training data are modeled as noisy measurements of a random function $f$ drawn from a GP prior, denoted $f \sim \mathcal{N}(0, k(\cdot, \cdot))$, where $k : \mathcal{X} \times \mathcal{X} \to \mathbb{R}$ is a kernel function over any domain $\mathcal{X}$. The observed training pairs $(\mathbf{x}_i, y_i)$ are collected as $X = [\mathbf{x}_1, \ldots, \mathbf{x}_T]$ and $\mathbf{y} = [y_1, \ldots, y_T] \in \mathbb{R}^T$, where, for an input $\mathbf{x}_i$, the observed value is modeled as $\mathbf{y}_i = f(\mathbf{x}_i) + \epsilon$, with $\epsilon_i \sim \mathcal{N}(0, \sigma^2)$. The kernel matrix on data is $K_X = [k(\mathbf{x}_i, \mathbf{x}_j)]_{i,j=1}^T \in \mathbb{R}^{T \times T}$. The posterior mean $\mu_{f|\mathcal{D}}$ and variance $\sigma_{f|\mathcal{D}}$ functions for GPs are:

$$\mu_{f|\mathcal{D}}(\mathbf{x}) := \mathbf{k}_{\mathbf{x}}^T \mathbf{z} \tag{34}$$
$$\sigma_{f|\mathcal{D}}(\mathbf{x}) := k(\mathbf{x}, \mathbf{x}) - \mathbf{k}_{\mathbf{x}}^T (K_X + \sigma^2 I)^{-1} \mathbf{k}_{\mathbf{x}} \tag{35}$$

where $\mathbf{k}_{\mathbf{x}} \in \mathbb{R}^T$ has as its $i^{th}$ entry $k(\mathbf{x}, \mathbf{x}_i)$, $\mathbf{z} = (K_X + \sigma^2 I)^{-1} \mathbf{y}$, and $I$ is an identity matrix. For GP regression on an arbitrary domain $\mathcal{X}$, the kernel function must be a p.d. kernel [23].

Naive approaches rely on the Cholesky decomposition of the matrix $K_X + \sigma^2 I$, which takes $\Theta(T^3)$ time [23]. To circumvent the $\Theta(T^3)$ runtime, recent works use iterative algorithms such as the conjugate gradient algorithm, which facilitate GP inference by exploiting fast kernel matrix-vector multiplication (MVM) algorithms, i.e., $\mathbf{v} \mapsto \mathbf{K_X v}$ [3]. In practice, these methods yield highly accurate approximations for GP posterior functions with a complexity of $\Theta(p \cdot T^2)$ for $p$ iterations of the conjugate gradient algorithm, as $\mathbf{mvm}(K_X) = T^2$, and $\mathbf{mvm}(M)$ is the operation count for multiplying matrix $M$ by a vector. $p \ll T$ proves to be efficient in practical application [3].

## B.2   Contextual GP Reward Model

Optimizing the $\mathcal{AF}$, i.e., UCB function, poses a significant challenge due to its enormous size of $\Pi^k$. Drawing inspiration from prior research on Bayesian optimization within combinatorial spaces, we employ a breadth-first local search (BFLS) to optimize the UCB acquisition function [2, 19]. The BFLS begins with the selection of several random top-k rankings. Subsequently, each specific top-k ranking is compared with the UCB values of its neighboring rankings, proceeding to the one with the highest UCB value.

The neighbors of a top-k ranking include all its permutations and the permutations of modified top-k rankings obtained by swapping one item with any of the remaining items. For any top-k ranking, there are $(n - k) \cdot k! + k!$ neighbors, which is often not huge as $k$ is often $\leq 6$. This search continues until no neighboring top-k ranking with a higher value is discovered. Although BFLS is a local search, the initial random selection and multiple restart points help it evade local minima, a strategy that previous studies have corroborated [19].

## B.3  Assessing GP-TopK Compute Requirements

> **Theorem 2.** *Assuming a fixed number of iterations required by the iterative algorithms, the total computational time for running the GP-TopK bandit algorithm for $T$ rounds of top-k recommendations, using the contextual product kernel (Equation 6), is $\mathcal{O}(k^2 c \ell T^2)$. This applies to WK, CK, and WCK top-k ranking kernels, where $\ell$ is the number of local search evaluations for selecting the next arm in every round.*

*Proof.* The proof can be straightforwardly derived by combining the results presented in Table 1, which succinctly summarizes the time complexities for each step of computing the UCB using both feature and kernel approaches. It is important to emphasize that iterative algorithms enhance results from $\mathcal{O}(T^4)$ to $\mathcal{O}(T^3)$ in computational complexity. Furthermore, these algorithms can further reduce complexity to $\mathcal{O}(T^2)$ when used with the feature approach.

The results presented in Table 1 can be validated through straightforward observations and by leveraging findings from previous Sections 2. Specifically, Section 2 offers proof for the $\mathbf{mvm}(K_X)$ row explicitly. For the *compute $K_{X_t}$* row, the complexity of kernel approaches is deduced from Algorithms 2 and 3. For feature approaches, the *compute $K_{X_t}$* row is inferred from the sparsity of the feature representations as stated in Claim 3. Lastly, the *memory* row is straightforwardly deduced for the kernel approach by counting its entries. For the feature approach, it is derived from the sparsity of the feature representations. $\qquad\square$

## B.4  Regret Analysis

In this section, we revisit Theorem 3 and provide its proof. The proofs build on the work by Krause et al. [14], delivering results for bounding the contextual regret in the context of the top-k ranking problem. To set the stage for our regret analysis, let's first define the critical term *maximum mutual information*, denoted by $\gamma_t$, is given below:

$$\gamma_t := \max_{X \subseteq \mathcal{X}: |X| = t} I(y_X; f), \qquad I(y_X; f) = H(y_X) - H(y_X | f),$$

where $I(y_X; f)$ quantifies the reduction in uncertainty (measured in terms of differential Shannon entropy) about $f$ achieved by revealing $y_A$ [27]. In Gaussian observation case, the entropy can be computed in closed form: $H(N(\mu, \Sigma)) = \frac{1}{2} \log |2 \pi e \Sigma|$, so that $I(y_X; f) = \frac{1}{2} \log |I + \xi^{-2} K_X|$, where $K_X = [k(x, x')]_{x, x' \in X}$ is the Gram matrix of $k$ evaluated on set $X \subseteq \mathcal{X}$. For the contextual bandit algorithm, $X$ represents contexts and arms considered until round $t$.

Before proving Theorem 3, we align the Krause et al. [14] results with our notation for consistency. Furthermore, we modify $\beta_t$ to accommodate embeddings encompassing negative values, aligning with the fact that contextual embeddings may exhibit negative dimensions.

> **Proposition 1** (Theorem 1, [14]). *Let $\delta \in (0, 1)$, and the unknown reward function $\hat{f}$ be sampled from the known GP prior with known noise variance $\sigma^2$. Suppose one of the following holds:*
>
> 1. *Assumption 1 holds and set $\beta_t = 2 \log(|\mathcal{X}| t^2 \pi^2 / 6 \delta)$.*
>
> 2. *Assumption 2 holds and set $\beta_t = 2B^2 + 300 \gamma_t \ln^3(t/\delta)$.*
>
> *Then the cumulative regret $\mathcal{R}_T$ of the contextual GP bandit algorithm with the UCB acquisition function is bounded by $\tilde{\mathcal{O}}(\sqrt{C_1 T \gamma_T \beta_T})$ w.h.p.    Precisely,*

$$\Pr\left\{ R_T \le \sqrt{C_1 T \gamma_T \beta_t} + 2 \quad \forall T \ge 1 \right\} \ge 1 - \delta, \text{ where, } C_1 = 8/\log(1 + \sigma^{-2}) \text{ and}$$
the notation $\tilde{\mathcal{O}}$ hides logarithmic factors in $n$, $\frac{1}{\delta}$ and $T$.

Proposition 1 shows that the regret $\mathcal{R}_T$ for the contextual GP bandit algorithm, utilizing the UCB acquisition function is bounded with high probability within $\tilde{\mathcal{O}}(\sqrt{C_1 T \gamma_T \beta_T})$, where the notation $\tilde{\mathcal{O}}$ hides logarithmic factors in $n$, $\frac{1}{\delta}$ and $T$. To ascertain the $\tilde{\mathcal{O}}$ order for $\mathcal{R}_T$, it is imperative to first bound the $\tilde{\mathcal{O}}$ order of $\gamma_T \beta_t$. We begin by examining the $\gamma_T$ term in the subsequent proposition.

> **Proposition 2.** *Under the assumptions of Theorem 3, $\gamma_T$ can be succinctly characterized as $\gamma_T = \mathcal{O}(n^2 c \log(n^2 T) + c \log T)$, which also simplifies to $\tilde{\mathcal{O}}(n^2 c)$, where the $\tilde{\mathcal{O}}$ notation omits logarithmic factors in $n$ and $T$.*

*Proof.* For the GP bandit algorithm with the UCB acquisition function, $\gamma_T = C \cdot \log\left(|I + \sigma^{-2} K_{X_T}|\right)$, where $C$ equals $(1/2) \cdot (1 - 1/e)^{-1}$ and $K_{X_T}$ represents the kernel matrix computed over contexts and arms across $T$ rounds [27, 14]. Precisely, $K_{X_T}$ is calculated using the contextual kernel defined in Equation 6. It is applied to contexts and top-k ratings from the feedback data $\mathcal{D}_t$, corresponding to Line 6 of the generic contextual bandit Algorithm 1.

Next, we leverage the characteristic of the contextual kernel being a product kernel. Consequently, the maximum mutual information term for the joint kernel, $\gamma_T$, can be upper bounded by $c \cdot (\gamma_T^\pi + \log T)$, where $c$ denotes the dimensionality of contexts and $\gamma_T^\pi$ represents the maximum information gain in a non-contextual setting [14]. Specifically, $\gamma_T^\pi$ is computed similarly but is confined to top-k rankings. That is, $\gamma_T^\pi = C \cdot \log\left(|I + \sigma^{-2} K_{X^\pi}|\right)$, with $K_{X_T^\pi}$ being calculated exclusively using the top-k kernels on the top-k rankings as selected by the bandit algorithm. $X_T^\pi$ represents the top-k rankings selected by the bandit algorithm, i.e., excluding the contexts from the collected feedback.

Recalling the formulation for top-k rankings kernels, we have $K_{X_T} = \Phi_{X_T^\pi}^T \Phi_{X_T^\pi}$, where $\Phi_{X^\pi} \in \mathbb{R}^{\binom{n}{2} \times T}$ comprises feature columns pertinent to the top-$k$ ranking kernels, as elucidated in Section A. Utilizing the Weinstein–Aronszajn identity, $\gamma_T^\pi$ is expressed as $C \cdot \log\left(|I + \sigma^{-2} \Phi_{X_T^\pi} \Phi_{X_T^\pi}^T|\right)$. Further, we deduce that $\gamma_T^\pi \le C \cdot \sum_{i=1}^{\binom{n}{2}} \log\left(|1 + \sigma^{-2} \lambda_i|\right)$, where $\lambda_i$ is an eigenvalue of $\Phi_{X_T^\pi} \Phi_{X_T^\pi}^T$. Given the Gershgorin circle theorem, which bounds all eigenvalues of a matrix by the maximum absolute sum of its rows, therefore we can conclude that $\gamma_T^\pi = \mathcal{O}(n^2 \log(n^2 T))$, as for all the columns of the $\Phi_{X^\pi}$ have bounded normed as given in Claims 2 and 3, i.e., $||\phi(\pi)||_2^2 \le 1$ [29].

By combining $\gamma_T^\pi = \mathcal{O}(n^2 \log(n^2 T))$ with the contextual product kernel, we obtain $\gamma_T = \mathcal{O}(n^2 c \log(n^2 T) + c \log T)$, thereby providing the claimed bound in the proposition. $\square$

Next, we build on Propositions 1 and 2 to prove the main theorem regarding the regret of the proposed GP-TopK bandit algorithm for top-k recommendations.

> **Theorem 3.** *If either Assumptions 1 or 2 hold, setting $\beta_t$ as $2\log\left(\frac{|\mathcal{C}| \cdot |\Pi^k| \cdot t^2 \cdot \pi^2}{6\delta}\right)$ and $300 \gamma_t \ln^3\left(\frac{t}{\delta}\right)$ respectively, the cumulative regret $\mathcal{R}_T$ of the GP-TopK bandit algorithm for top-k recommendations can, with at least $1 - \delta$ probability, be bounded by $\tilde{\mathcal{O}}(n\sqrt{C_1 T c(\log|\mathcal{C}| + k + \log(T^2 \pi^2/6\delta))})$ under Assumption 1, and $\tilde{\mathcal{O}}(n\sqrt{C_1(2B^2 c + 300 n^2 c^2 \ln^3(T/\delta))T})$ under Assumption 2. Here, $C_1 = \frac{8}{\log(1+\xi^{-2})}$, and $\tilde{\mathcal{O}}$ excludes logarithmic factors related to $n$, $k$, and $T$.*

*Proof.* We will prove the above theorem for both cases separately.

**For Assumption-1.** Given $|\mathcal{C}|$ is finite and $\beta_T = 2\log(|\mathcal{D}| T^2 \pi^2 / 6\delta)$. First, we focus on bounding $\beta_T$ as follows:

$$\beta_T = 2\log(|\mathcal{D}| T^2 \pi^2 / 6\delta)$$

$$= \mathcal{O}\left(\log|\mathcal{C}| + \log|\Pi^k| + \log(T^2\pi^2/6\delta)\right)$$

As $\binom{n}{k} \leq n^k$ and $k! \leq k^k$, we also have $\log|\Pi^k| = \log\left(\binom{n}{k}k!\right) \leq \log\left(n^k k^k\right) = \mathcal{O}(k\log(nk))$, which implies that $\beta_T = \mathcal{O}(\log|\mathcal{C}| + k\log(nk) + \log(T^2\pi^2/6\delta))$. Combining this with Proposition 2, we have following:

$$\mathcal{O}(\gamma_T\beta_T) = \mathcal{O}\left((n^2 c\log(n^2 T) + c\log T)(\log|\mathcal{C}| + k\log(nk) + \log(T^2\pi^2/6\delta))\right)$$
$$= \mathcal{O}\left(n^2 c\log(n^2 T)(\log|\mathcal{C}| + k\log(nk) + \log(T^2\pi^2/6\delta))\right) \quad \text{(Ignoring } c\log T \text{ term)}$$
$$= \tilde{\mathcal{O}}\left(n^2 c\left(\log|\mathcal{C}| + k + \log(T^2\pi^2/6\delta)\right)\right).$$

Thus, we showcase the asserted bound for the regret $\mathcal{R}_T$ as $\tilde{\mathcal{O}}\left(\sqrt{C_1 T\gamma_T\beta_T}\right) = \tilde{\mathcal{O}}\left(n\sqrt{C_1 Tc(\log|\mathcal{C}| + k + \log(T^2\pi^2/6\delta))}\right)$.

**For Assumption-2.** Given $\|f\|_k \leq B$ and $\beta_t = 2B^2 + 300\gamma_t\ln^3(t/\delta)$. First, we bound the $\beta_T$ term using Proposition 2 as follows:

$$\beta_T = 2B^2 + 300 \cdot \gamma_T \cdot \ln^3(T/\delta),$$
$$= 2B^2 + 300 \cdot \left(n^2 c\log(n^2 T) + c\log T\right) \cdot \ln^3(T/\delta).$$

Using the above result, we have the following:

$$\mathcal{O}(\sqrt{C_1 T\gamma_T\beta_T}) = \mathcal{O}\left(\sqrt{C_1 T\gamma_T \cdot \left(2B^2 + 300 \cdot \gamma_T \cdot \ln^3(T/\delta)\right)}\right),$$
$$= \mathcal{O}\left(\sqrt{C_1 Tn^2 c\log(n^2 T) \cdot \left(2B^2 + 300 \cdot n^2 c\log(n^2 T) \cdot \ln^3(T/\delta)\right)}\right),$$
$$= \tilde{\mathcal{O}}\left(n\sqrt{C_1 Tc(2B^2 + 300n^2 c\ln^3(T/\delta))}\right).$$

$\square$

**Comparison with Srinivas et al. (2010).** Using the identity kernel for top-k rankings, we can develop a finite-dimensional feature for the contextual kernel and apply Theorem 5 by Srinivas et al. (2010). Given that $\gamma_T = O(n^k c\log T)$, the regret bounds are as follows under both assumptions. For instance, the calculations for the $\mathcal{O}(\sqrt{C_1 T\gamma_T\beta_T})$ under the Assumption 2 are as follows:

$$\mathcal{O}(\sqrt{C_1 T\gamma_T\beta_T}) = \mathcal{O}\left(\sqrt{C_1 T\gamma_T \cdot \left(2B^2 + 300 \cdot \gamma_T \cdot \ln^3(T/\delta)\right)}\right),$$
$$= \mathcal{O}\left(\sqrt{C_1 T\left(n^k c\log T\right) \cdot \left(2B^2 + 300 \cdot (n^k c\log T) \cdot \ln^3(T/\delta)\right)}\right),$$
$$= \tilde{\mathcal{O}}\left(n^{\frac{k}{2}}\sqrt{C_1 Tc(2B^2 + 300n^k c\ln^3(T/\delta))}\right).$$

Similarly, we can analogously perform the analysis for Assumption 1 and combine it with Proposition 1 to obtain the regret bounds mentioned in the Table 3.

## C Experiments – Omitted Details

This section presents omitted details from the main body of the text.

### C.1 Compute resources

We utilized multiple NVIDIA Tesla M40 GPUs with 40 GB RAM on our in-house cluster for our experiments. The experiments in Section 5 required approximately 5 GPU-hours for small arm space and 24 GPU-hours per iteration for large arm space. We conducted about 50 to 100 iterations throughout the project. The results reported in Section C.3 required the same computational resources as the large arm space experiments.

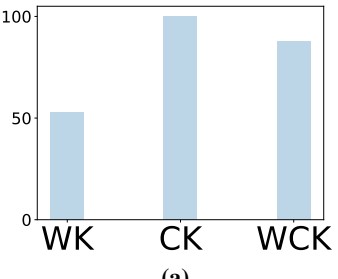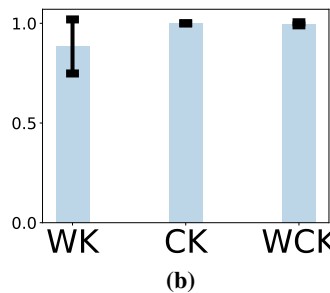

|     |     |
| :-: | :-: |
| **(a)** | **(b)** |

**Figure 4:** Local search results for optimizing combinatorial objectives in $\Pi^k$ for $n = 50$ and $k = 6$. For details, see the textual description. Left (a) shows how many times out of 100 trials the local search recovers the exact maximizer, i.e., $\pi^{'}$, and right plot (b) shows the average value of the objective for the returned maximizer. These results indicate that the local search utilized in this work is effective.

## C.2 Bandit Simulation and Hyper-parameter Configurations – Omitted Details

To set up the simulation, we utilized embeddings trained on the MovieLens dataset using a collaborative filtering approach [6]. We consider a $1M$ variant of the MovieLens dataset, which contains 1 million ratings from 6040 users for 3677 items. Specifically, we train user embeddings $\mathbf{c}_u$ and item embeddings $\theta_i$ such that the user's attraction to the items are captured by the inner product of the user embedding with the item embeddings, respectively. Both context and item embeddings, i.e., $\mathbf{c}_u$ and $\theta_i$, are 5-dimensional, optimized by considering the 5-fold performance on this dataset. The reward provided in our experiments is contaminated with zero mean and standard deviation equals 0.05.

For the $\epsilon$-*greedy* baselines, we considered various values of $\epsilon$ are considered, specifically $\epsilon = \{0.01, 0.05, 0.1\}$. The outcomes are presented for the configuration that demonstrates optimal performance. For *MAB-UCB* baseline, the algorithm has an upper confidence score $ucb(i) = \overline{\mu}_i + \beta_{mab}\sqrt{\frac{2\ln(t+1)}{n_i}}$ [11]. Here, $\overline{\mu}_i$ represents the average reward, $n$ denotes the total number of rounds, and $n_i$ signifies the frequency of arm $i$ being played. $\beta_{mab}$ is a hyper-parameter. We evaluate $\beta_{mab}$ values within the set $\{0.1, 0.25, 0.5\}$ and disclose results for the best-performing configuration. For the parameters of proposed GP-TopK bandit algorithms, we set $\beta_t = \beta_{gp} \cdot \log(|\mathcal{X}| \cdot t^2 \cdot \pi^2)$ with $\beta_{gp} \in \{0.05, 0.1, 0.5\}$, reporting results the value that yields the best performance. The choice of $\beta_t$ is informed by prior work in GP bandits [27]. The selection of $\sigma$ for all variants is determined by optimizing the log-likelihood of the observed after every 10 rounds by considering values in the set $\{0.01, 0.05, 0.1\}$.

## C.3 Additional results

**Local search** results for optimizing combinatorial objectives in $\Pi^k$ for $n = 50$ and $k = 6$. Specifically, $\pi^{\star} = \max_{\pi} \phi^r(\pi)^T \phi^r(\pi^{'})$, where $\phi^r(\pi^{'}$ represents the feature vector for Kendall kernels on top-k rankings. Notably, for this optimization problem, it is known that the optimal value is 1 obtained by only $\pi^{'}$. Figure 4 shows results for this optimization problem when applied to WK, CK, and WCK kernels.

**Reward results** for large arm space for the nDCG + diversity reward. Similar to Figure 3, a large setup with $n = 50$ for $k = 3$ and $k = 6$, is considered. For $k = 6$, the possible arms are over $1.1 \times 10^{10}$ possible top-k rankings. Given the vastness of this arm space, computing the optimal arm for the diversity reward is not straightforward. Therefore, we focus on reporting the cumulative reward in Figure 5. We implement this setup using a Local search in batch mode, updating every 5 round and considering a substantial horizon of $T = 100$ rounds. Specifically, we use 5 restarts, 5 steps in every search direction, and start with 1000 initial candidates. Figure 5 shows that the WCK approach demonstrates superior performance, continuing to learn effectively even after extensive rounds.

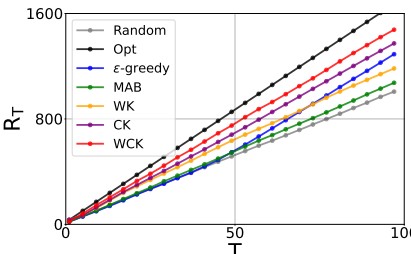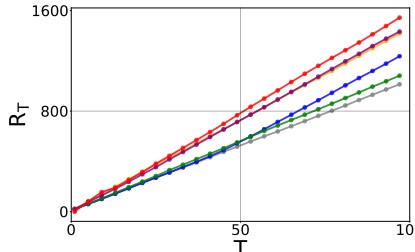

**Figure 5:** Comparative evaluation of bandit algorithms for large arm spaces for the nDCG + diversity reward, with $> 1.1 \times 10^5$ for the left plot and $> 1.1 \times 10^{10}$ for the right plot, respectively. Cumulative reward with respect to the rounds of the bandit algorithm is depicted. Results are averaged over 6 trials. In both settings, the WCK approach outperforms other baselines. For more details, see the textual description.

