# OpenReview forum: "Gaussian Process Bandits for Top-k Recommendations"
_NeurIPS.cc/2024/Conference — NeurIPS 2024 poster_

### Official Review · Reviewer_N1XB · 2024-07-03

**Soundness:** 3
**Presentation:** 3
**Contribution:** 2
**Rating:** 6
**Confidence:** 3

**Summary:**

This paper addresses the problem of top-k recommendation with bandit feedback, where the goal is to recommend a list of k items and receive feedback only on the overall quality of the list, not individual items. This captures the notion that user satisfaction depends on the overall quality of the recommendations, not just the single most relevant item. To tackle this, the authors propose a novel algorithm combining Gaussian Process Upper Confidence Bound (GP-UCB) with a specifically designed weighted Kendall kernel. The paper proposes faster inference exploiting features of the Kendall kernel, improving computational complexity from $O(T^4)$ to $O(T^2)$. An upper bound on cumulative regret is derived, showing sublinear regret. Finally, synthetic experiments show that the proposed algorithm can outperform a few baselines.

Disclosure: I have reviewed a previous version of this paper.

**Strengths:**

Strengths:
* Extends the literature on top-k recommendations with a GP-UCB algorithm
* Well presented

**Weaknesses:**

Weaknesses:
* The approach does not seem scalable to real-world problems
* The theoretical contribution could be further clarified

**Questions:**

* The theoretical analysis builds on the analysis of Krause & Ong (2011), adapted to the Kendall kernel. Can the authors please clarify the technical contribution (novelty) compared to Krause & Ong (2011).
* The approach does not seem practical. The empirical evaluation is done with a small number of items (n<=50) and short horizon (T<500). In real-world applications the number of items is often much larger and the number of rounds can also be large.
* The acquisition function is not applied to all possible top-k rankings, rather a local search approach is used. The regret analysis does not account for the approximation due to the local search. The implementation also uses an approximate posterior rather than an exact posterior. This creates a discrepancy between the implementation of the algorithm and the theoretical analysis. The authors are somewhat upfront about this.
* L304 (Evaluation for large arm space):
First, I wouldn’t say that n=50 is large. In real-world applications n is in the millions (e.g., music, books) or billions (e.g., videos).
Second, when showing the regret in Figrue 3 you should actually show all iterations (1-500), not just the ones that include a model update (every 5 steps).

* A few comments from the previous review that are still not fixed:
  * You still have an overloaded notation $\sigma$ for both a ranking (L88) and the sigmoid function (L273). In the previous version you had a three-way overload of $\sigma$ and now it’s only two, but this is easy to fix.
  * Another example from the rebuttal: “CG is the conjugate gradient algorithm. We will clarify this in the future.”
In the current version this part is moved to the appendix (L707), but still appears as “CG”, without clarification.
  * L283: “We set $\lambda = 0.25$ to emphasize relevance over diversity.” This value actually gives a 0.25 weight to ndcg and a 0.75 weight to diversity so the weight for diversity is higher. I also mentioned this in my review to the previous version, did I not understand this correctly and this is why you ignored the comment?

**Limitations:**

No societal issues.

---

> ### Author Rebuttal · Authors · 2024-08-06
>
> **Dear Reviewer N1XB,**
>
>
> We are thankful for your continued positive outlook on this work and encouraged to find recognition of our contributions. Thanks for identifying the typos; we will fix them. Next, we respond to your questions and weaknesses below.
>
> > clarify the technical contribution (novelty) compared to Krause & Ong (2011).
>
> We utilize Theorem 1 from Krause & Ong (2011) (presented as Proposition 1), which relates the regret bound to the maximum mutual information, generalizing the contextual GP-UCB bandits from Srinivas et al. (2010). Building on Proposition 1, we focus on bounding the maximum mutual information by leveraging Weinstein–Aronszajn identity and the feature representation of top-k rankings kernels introduced in Section 3 (as given in Proposition 2). This result in Proposition 2 is not and yields significantly tighter regret bounds, as shown in Table 3, following your previous review suggestion. For further details, please see Section B.4.
>
> > The approach does not seem practical, and the empirical evaluation is done with a small number of items (n<=50) and a short horizon (T<500).
>
> The empirical assessment was designed to demonstrate the effectiveness of our approach in terms of regret minimization, focusing on a large arm space setting (e.g., 10+ billion arms for n=50 and k=3). The chosen parameters of n and T were intended to provide a conclusive comparison of the regret minimization effectiveness against other baseline algorithms rather than to showcase scalability to "practical" real-world scales.
>
> Regarding the application in practical scenarios, we recognize that real-world systems often deal with billions of items, far exceeding the 50 items considered in our study. However, practical recommendation systems typically employ a multi-staged architecture involving rankers and re-rankers that progressively filter and narrow down the item pool from billions to often fewer than 50 items (see [1] and [2]). Bandit algorithms, or Bayesian optimization techniques, are usually applied at the final stage of these systems, suggesting that our approach could be more applicable to practical settings than it might seem.
>
> Our theoretical analysis indicates that the proposed algorithm is expected to yield further improvements for larger horizons, i.e., for $T>500$, the approach shall scale much better.
> It's important to note that the primary aim of this work is not to deliver a ready-to-deploy industrial-scale recommendation bandit algorithm but rather to advance research in this area under a more generalized setting with fewer assumptions. Nevertheless, we are open to refining our approach based on your feedback to better align with practical demands and real-world applicability.
>
> * [1]. LinkedIn blog on building large-scale system
> https://www.linkedin.com/blog/engineering/recommendations/building-a-large-scale-recommendation-system-people-you-may-know
> * [2]. IJCAI tutorial on Bayesian Optimization for Balancing Metrics in Recommender Systems https://ijcai20.org/t03/
>
> > The acquisition function is not applied to all possible top-k rankings, and the regret analysis does not account for the approximation
>
> In principle, the local search can explore all possible top-k rankings. Still, we acknowledge that it may not always yield a global optimizer. Addressing this approximation challenge is notably complex. Most literature, including the foundational GP-UCB paper by Srinivas et al. (2010), which received the ICML-2020 Test of Time Award, assumes exact optimization of the acquisition function and proves regret bounds under this assumption. The gap between theory and practical optimization remains underexplored. The only related study we are aware of is by Jungtaek Kim and collaborators in their paper "On Local Optimizers of Acquisition Functions in Bayesian Optimization," which investigates this issue in continuous input spaces and presents challenging results under several assumptions about the behavior of the reward function,  http://mlg.postech.ac.kr/~jtkim/papers/ecmlpkdd_2020.pdf.
>
> > Figure 3 should show all iterations (i.e., in the batch mode)
>
> Thanks for raising this point. The added PDF provides the regret computation for all iterations. It yields higher variance and slightly more regret while keeping the conclusions and observations consistent, as we anticipated and overserved in our experiments earlier.
>
>
> **We value your detailed feedback and welcome further engagement to strengthen this work.**

---

> > ### Comment · Reviewer_N1XB · 2024-08-13
> >
> > Thanks for the clarifications.

---

### Official Review · Reviewer_DrUM · 2024-07-12

**Soundness:** 2
**Presentation:** 3
**Contribution:** 3
**Rating:** 5
**Confidence:** 3

**Summary:**

The paper introduces a contextual bandit algorithm for top-k recommendations, leveraging Gaussian processes with a Kendall kernel to model the reward function. The proposed method utilizes full-bandit feedback without assumptions such as semi-bandit feedback or cascade browsing. Theoretical analysis demonstrates a sublinear regret with the number of rounds and arms, and empirical results from simulations show superior performance compared to other baselines. The authors also improve the computational efficiency and memory requirements of the algorithm.

**Strengths:**

1. The paper is well-organized and well-written, with clear explanations and detailed experimental results.
2. The proposed algorithm achieves state-of-the-art performance in top-k recommendation tasks compared with various baselines. The theoretical analysis provides a solid foundation for the algorithm's performance guarantees.
3. The improvements in computational efficiency and memory requirements are novel, and technical, making the algorithm more practical for real-world applications.

**Weaknesses:**

1. The paper claims that existing bandit algorithms impose strict assumptions about feedback models, e.g., semi-bandit feedback or cascade browsing. I agree with this statement. However, the paper addresses this problem by directly using Gaussian processes to model the reward function. Isn't this also a restrictive assumption? The authors should clarify this point.
2. The paper lacks a detailed discussion on the choice of the Kendall kernel. The authors should provide more insights into why this kernel was chosen and its advantages over other kernels.
3. There are existing works about top-k combinatorial bandits with full-bandit feedback, such as Rejwan and Mansour (2020) [1]. However, the paper does not compare its method with these existing works. The authors should compare their method with these existing works to demonstrate the difference and novelty of their approach.

[1] Rejwan, Idan, and Yishay Mansour. "Top-$ k $ combinatorial bandits with full-bandit feedback." Algorithmic Learning Theory. PMLR, 2020.

**Questions:**

Questions:
1. Why did you choose the Kendall kernel for the Gaussian process? What are the advantages of this kernel over other kernels?
2. How does your algorithm's performance compare to existing works that use full-bandit feedback, such as Rejwan and Mansour (2020)?

**Limitations:**

Limitations are adequately discussed in the paper.

---

> ### Author Rebuttal · Authors · 2024-08-06
>
> **Dear Reviewer DrUM,**
>
>
> We appreciate your positive feedback on multiple aspects, including writing, results, theoretical analysis, and novel improvements in computational efficiency and memory requirements. We respond to your questions below.
>
> > Are Gaussian processes restrictive for modeling rewards?
>
> The ability of GPs to accurately model a specific reward function depends on the expressivity of the RKHS associated with the utilized kernel. Informally, as long as there exists a vector 𝑤, s.t, the norm $||𝑤^T𝜙(𝜋)−𝑓||$ is small, where 𝑓 is the true reward function and \phi(\pi) is the feature space, the proposed algorithm should be effective. This statement can be extended to contextual setup as well. It’s worth noting that Assumption 2 outlines the same principle, similar to prior works such as Krause & Ong (2011). Following your suggestion, we assessed the accuracy of the RKHS of the proposed kernel in approximating reward signals through linear regression in the feature space. We trained on 10,000 arm configurations and tested on another 10,000, repeating this six times to measure MSE performance for the nDCG reward as detailed in the paper. The MSE for the NDCG was 0.816 +/- 0.017 for WK, 0.157 +/- 0.009 for CK, and 0.069 +/- 0.010 for WCK.
>
> > Could you provide a Detailed discussion on the choice of the Kendall kernel? Why did you choose it for the Gaussian process? What are its advantages over other kernels?
>
> Section 2.1 discusses the choice of the Kendall kernel, emphasizing its suitability for Gaussian processes due to being positive definite, right-invariant, and appropriate for rankings. It stands out as one of the few kernels that meet these requirements, alongside the Mellow kernel, built on top of the Kendall kernel. We opted for the Kendall kernel for its simplicity and potential ease of adaptation to top-k rankings. Considering Mellow Kernels would be an exciting avenue for future work. Section 2.2 elaborates on the appropriateness of the Kendal kernel, with examples provided in Table 2.
>
> > Compared to existing works that use full-bandit feedback, such as Rejwan and Mansour (2020)?
> Rejwan and Mansour (2020) present a study on full-bandit feedback for combinatorial bandits. They focus on selecting a subset of top-k items without concern for their arrangement, which is a different problem setting than what this work considers.
>
> **We appreciate your valuable feedback and look forward to learning from it to enhance this work.**

---

> > ### Comment · Reviewer_DrUM · 2024-08-10
> >
> > Thanks for your responses and I will keep my rating. I also recommend adding a 'Related Work' section to help readers better understand the distinctions between your work and other related research.

---

> > > ### Author Response · Authors · 2024-08-12
> > > **Thanks for acknowledging the comment.**
> > >
> > > Thank you, Reviewer DrUM, for your prompt response. We genuinely appreciate it. We’ll consider adding a related work section in the appendix due to space constraints if it’s not feasible in the main section.
> > >
> > > At your convenience, we would greatly appreciate it if you could clarify whether your concerns have been addressed or not. While your clarification is very important to us and is a crucial step in the review process, we understand if you’re unable to reply.
> > >
> > > Thanks again!

---

> > > > ### Comment · Reviewer_DrUM · 2024-08-13
> > > >
> > > > Thank you and my concerns have been addressed. Regarding my first question, I realize now that my initial confusion stemmed from the statement: "This necessitates algorithms for full-bandit feedback settings **without assumptions about the objective or reward**." in Lines 40-41. This sentence appears contradictory because using Gaussian processes to model the reward function is still an assumption for me. I suggest that the authors consider rephrasing this sentence for clarity.
> > > >
> > > > I will maintain my current rating, considering a balanced comparison with other papers in my review batch.

---

### Official Review · Reviewer_9Xg2 · 2024-07-13

**Soundness:** 3
**Presentation:** 4
**Contribution:** 2
**Rating:** 5
**Confidence:** 3

**Summary:**

The authors propose a new bandit algorithm for top-k recommendations that utilizes a Gaussian Process to model the reward of the exponentially large set of arms, and provide regret bounds which improve upon the naive approach of modeling each arm independently. The authors also present a new kernel and show that the associated kernel matrix displays a particular structure that can be exploited to efficiently compute GP predictions. Finally, experimental validation show the improved regret with respect to baseline algorithms and other similar kernels.

**Strengths:**

- The paper is very well presented and easy to read, and the proofs are also very clear.
The authors provide meaningful insight into the strengths of their new kernel through small examples and the results of the experiments.
- The computational costs of a Gaussian Process using the new kernel are extensively addressed, and their approach is significantly more efficient than a standard procedure.

**Weaknesses:**

- The authors claim the new algorithm to be their primary contribution, though it is a simple adaptation of a contextual GP-UCB to top-k recommendations, and a lot of attention is dedicated instead to the new kernel and its computational costs.
- Despite the strengths of the new kernel, it is not particularly novel as it is a straightforward combination of existing kernels.
- Figures in the Experiments section should be more clear about the confidence ranges (the confidence level in Figure 2 is not mentioned, and the ranges appear to be missing entirely in Figures 3 and 5).

I found a few typos and confusing passages, but overall, they didn't affect the reading experience. I will report them here as a feedback:
line 99: typo ww(...) -> w(...)
lines 254-256: this sentence sounds like the new terms involving n in the bounds are n^{k/2-1} and n^{k-1}, maybe the intended meaning was to highlight the improvement with respect to the cited results?
lines 281-283: \lambda = 0.25 emphasizes diversity over relevance
line 751: typo shoes -> shows
line 821: typo very -> every (also possibly missing a subject after "observed"?)

**Questions:**

1) The number of local searches is a substantial factor in the overall time complexity of the proposed algorithm, and the number of local searches seems to be particularly high for the large arm space scenarios. Since optimizing the computational costs of the algorithm is a big part of this work, I would like to ask the authors whether they have considered and whether it is possible and/or practically useful to further exploit the structure of the kernel to avoid recomputing the predictor entirely when moving between neighbors.

---

> ### Author Rebuttal · Authors · 2024-08-06
>
> **Dear Reviewer 9Xg2,**
>
> We appreciate your positive feedback on the readability and clarity of the proofs. We respond to your comments below. Thanks for pointing out typos and flipped values of \lambda parameter, i.e., when we wrote \lambda = 0.25, we meant to write \lambda = 0.75.
>
> >  new algorithm to be their primary contribution
>
> We agree that our primary contribution can be viewed as an efficient implementation of the GP-UCB for the ranking kernels studied—this is a significant and essential part of our work. At the same time, although we do not alter the schematic of the GP-UCB, our contributions extend beyond the efficient implementation of the kernels, novel kernel for top-k rankings, exploring the full-bandit feedback settings, empirically showing optimal arm selection for the contextual acquisition function over the top-k domain, and providing its regret analysis.
>
> >  more clear about the confidence ranges
>
> Thank you for your critical observation. The confidence level for Figure 2 is 95%. In Figure 5, confidence intervals were omitted due to poor visibility, while Figure 3 exhibits confidence behavior similar to that of Figure 2. We have included these figures in the attached pdf for your reference.
>
> >  on the possibility of efficient local search exploiting the kernel structure
>
> Thank you for your insightful question. The number of local searches significantly influences our algorithm, which grows linearly with the number of items in our implementation due to the neighbors considered by the local search. There might be a strategy to improve this further to reduce the number of neighbors.  However,  it does not seem trivial to create smaller and more relevant neighborhoods for the local search by leveraging the relationship between the UCB acquisition function and the feature representation of the arm, i.e., $\phi(\pi)^T{w}_t + \beta \sqrt{\phi(\pi)^T M_t \phi(\pi)}$, where ${w}_t$ and $M_t$ are fixed matrices at the t-th step.
>
> **We are grateful for the valuable feedback on our work. We welcome further engagement and look forward to learning from your perspective.**

---

> > ### Comment · Reviewer_9Xg2 · 2024-08-13
> >
> > Thanks for the clarifications. Thanks for your responses and I will keep my rating.

---

### Official Review · Reviewer_tLdn · 2024-07-20

**Soundness:** 4
**Presentation:** 4
**Contribution:** 4
**Rating:** 8
**Confidence:** 3

**Summary:**

This paper considers the slate recommendation problem where k items are simultaneously recommended to a user at the same time (in a banner or "slate").  In order to solve this problem the authors adapt existing Gaussian process methods to the top-k setting by modifying Kendall kernels to the top-k setting.  They also introduce some computational tricks that allow for the algorithm to be quadratic in the embedding size.  They also provide a regret analysis and empirical studies to defend their algorithm.

**Strengths:**

The paper is extremely readable and well presented, I particularly like the boxes highlighting key claims and the examples e.g. Table 2 and Figure 1.  The mathematical notation is clean and consistently used, and the authors are clearly in command of the literature they are building on.  This is particularly evident when the authors are able to draw a clean distinction between existing work and their contributions.

The experiments are well designed, cleanly implemented, well executed and presented immaculately.

**Weaknesses:**

I would like to provide some commentary on this paper and how it might relate to production recommender systems without in any way wanting to diminish the excellent work done in this paper.  NeurIPS is a primarily academic research venue and this paper is very well executed and it is very pleasing to see a recommender systems paper of such quality.

I would however be very surprised if this method or a method derived from it was applied in a real system for a number of reasons.

First, I like the authors use of Figure 1 in order to highlight limitations of the cascade model (the even simpler position based model has similar problems), there have however been models proposed for this case including the Probabilistic Rank and Reward model - https://arxiv.org/pdf/2208.06263 .  The "full bandit framework" while useful does discard the useful preference information about what was clicked (as well as saying the banner was clicked somewhere), this information is usually quite valuable.

The covariance functions of the model (equation 6) is very well explained and the idea is interesting.  A further useful note is that using this multiplied form the reward is correlated if both the context is similar _and_ the actions (top-k) are similar, due to multiplication if either is approximately zero the covariance is zero.  The covariance in the action space is rather limited however as it has no notion for some actions being similar.  For example if you are going to recommend action movies then you might fill the top-k with totally different action movies yet still have a correlated reward.  The model in this paper has some similarities, including the factorized covariance matrix https://arxiv.org/abs/2008.12504

Finally, this paper learns a unique reward for every top-k ordering.  This results in a combinatorial explosion both in learning and in delivering recommendations.  While the assumption is interesting it does not allow for finding the best recommendations performing an argsort often approximated using a fast maximum inner product search (e.g. LSH or HNSW), it seems difficult to relax these assumptions in real world systems.

**Questions:**

Do you agree with the limitations discussed above?

**Limitations:**

yes

---

> ### Author Rebuttal · Authors · 2024-08-06
>
> **Dear Reviewer tLdn,**
>
> We appreciate your positive feedback on the readability and design of our experiments. We address your comments in the order they were presented.
>
> > On the cascade model, utilize click information.
>
> We acknowledge that exploiting click information can be valuable. Still, it can also be misleading when the cascade model or item presentation order is ambiguous, e.g., in the scenarios given in the paper. Our approach addresses scenarios where the cascade model may not be appropriate. We certainly agree that click information is valuable, and it’s worth exploring a middle ground between full-bandit feedback and more restricted models like the cascade model.
>
>
> > On covariance functions and its shortcomings.
>
> Thank you for highlighting the limitations of the proposed product kernel, which requires both context and top-k rankings to be similar to utilize data from previous rounds. This limitation can be mitigated using other kernels, such as the additive kernel (similar to Krause et al. [14]). The contributions to accelerate the bandit algorithm remain applicable even with the additive kernel. We will clarify this further in the final draft. Exploring even broader classes of kernels for ranking/context spaces that admit efficient algorithms could be an exciting direction for future work.
>
> > Finding recommendations with inner product search
>
> We agree that the proposed approach does not allow for straightforward inner product search and requires an optimization algorithm due to the more generic setting, i.e., non-composability of rewards over items. This is also true for the GP bandit algorithm given by Wang et al. [32], despite having a much more relaxed scenario of the semi-bandit feedback.
>
> **Thank you for the insightful review!**

---

### Author Rebuttal · Authors · 2024-08-06

**Dear Program Chairs, Area Chairs, and Reviewers,**

We appreciate the constructive feedback on our work. It is encouraging that all reviewers affirm this work's soundness, presentation, and contributions. Below, we address the questions and concerns raised by each reviewer. We provide additional figures as reviewers  9Xg2 and DrUM requested in the attached single-page pdf.

**Thanks again!**

---

### Decision · Program_Chairs · 2024-09-25

**Decision:**

Accept (poster)

**Comment:**

This paper considers the slate recommendation problem where $k$ items are simultaneously recommended to a user at the same time. To solve this problem the authors adapt existing Gaussian process methods to the top-$k$ setting by modifying Kendall kernels to the top-$k$ setting. They also provide a regret analysis and empirical studies to justify their proposed algorithm.

Based on the reviews and ratings, this paper is borderline, but more on the accept side. All reviewers recommend to accept this paper, and they all agree that the paper is well written and easy to read. However, while this paper clearly has many strengths, it also has some weaknesses, as pointed out by the reviewers. In particular, I agree with Reviewer 9Xg2 that the novelties of the proposed algorithm seem to be limited.